# IN-CONTEXT LEARNING DYNAMICS WITH RANDOM BINARY SEQUENCES

**Eric J. Bigelow**[1,2] [*], **Ekdeep Singh Lubana**[2,3],
**Robert P. Dick**[3], **Hidenori Tanaka**[2,4] [†], **Tomer D. Ullman**[1,2] [†]

[1]Psychology Department, Harvard University, Cambridge, MA, USA
[2]Center for Brain Science, Harvard University, Cambridge, MA, USA
[3]EECS Department, University of Michigan, Ann Arbor, MI, USA
[4]Physics & Informatics Laboratories, NTT Research, Inc., Sunnyvale, CA, USA

## ABSTRACT

Large language models (LLMs) trained on huge text datasets demonstrate intriguing capabilities, achieving state-of-the-art performance on tasks they were not explicitly trained for. The precise nature of LLM capabilities is often mysterious, and different prompts can elicit different capabilities through in-context learning. We propose a framework that enables us to analyze in-context learning dynamics to understand latent concepts underlying LLMs' behavioral patterns. This provides a more nuanced understanding than success-or-failure evaluation benchmarks, but does not require observing internal activations as a mechanistic interpretation of circuits would. Inspired by the cognitive science of human randomness perception, we use random binary sequences as context and study dynamics of in-context learning by manipulating properties of context data, such as sequence length. In the latest GPT-3.5+ models, we find emergent abilities to generate seemingly random numbers and learn basic formal languages, with striking in-context learning dynamics where model outputs transition sharply from seemingly random behaviors to deterministic repetition.

## 1 INTRODUCTION

Large language models (LLMs) achieve high performance on a wide variety of tasks they were not explicitly trained for, demonstrating complex, emergent capabilities (Brown et al., 2020; Radford et al., 2019; Wei et al., 2022a; 2023; 2022b; Lu et al., 2023; Bommasani et al., 2021; Wei et al., 2021; Kojima et al., 2022; Pan et al., 2023; Pallagani et al., 2023; Li et al., 2022). In-context learning (ICL) describes how task-specific patterns of behavior are elicited in LLMs by different prompts (or *contexts*) (Brown et al., 2020; Wei et al., 2022b; Dong et al., 2022; Zhou et al., 2022; Wang et al., 2023; Shin et al., 2022; Min et al., 2022; Si et al., 2023; Xie et al., 2022; Zhang et al., 2023b). Although no weight updates occur in ICL, different input contexts can activate, or re-weight, different latent algorithms in an LLM, analogous to updating model parameters to learn representations in traditional gradient descent learning methods (Goldblum et al., 2023; Dai et al., 2023; Ahn et al., 2023; Von Oswald et al., 2023; Akyürek et al., 2022; Li et al., 2023a). However, two seemingly equivalent prompts can evoke very different behaviors in LLMs (Si et al., 2023). ICL makes interpreting what representations in an LLM are being evoked by a given input more challenging than with task-specific language models, where the space of capabilities is more restricted and well-defined.

Our central motivation is to interpret latent *concepts* underlying complex behaviors in LLMs by analyzing in-context learning behavioral dynamics, without directly observing hidden unit activations or re-training models on varied datasets. Inspired by computational approaches to human cognition (Tenenbaum, 1998; Griffiths & Tenenbaum, 2003; Piantadosi et al., 2012; Spivey et al., 2009), we model and interpret latent concepts evoked in LLMs by different contexts without observing or probing model internals. This approach, which we call *Cognitive Interpretability*, is a middle ground between shallow test-set evaluation benchmarks on one hand (Srivastava et al., 2022; Saparov & He, 2022; Min et al., 2022; Lanham et al., 2023; Ganguli et al., 2023; Bowman et al., 2022; Kadavath

---

[*]Correspondence to: `ebigelow@g.harvard.edu`
[†]Equal contribution

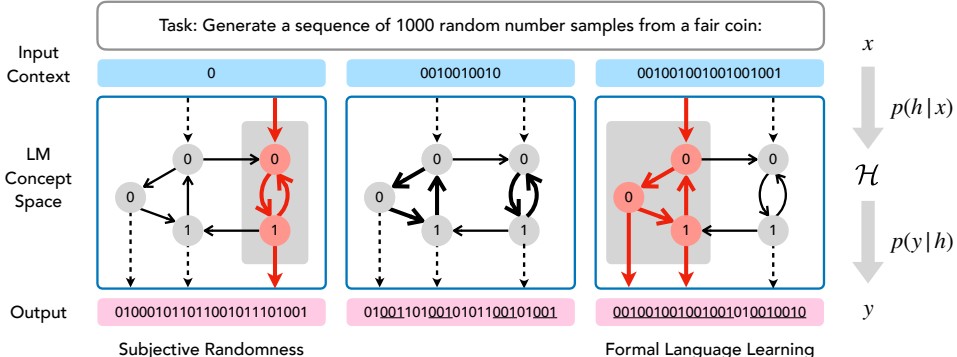

Figure 1: **Overview of our modeling framework.** Given a pre-trained Large Language Model, we systematically vary input context prompts, in this case $x = (001)^n$ with $n = \{2, 4\}$ (`001001`, `001001001001`). LLM outputs $y$ (Bottom) significantly as a function of $x$ (Top), based on some unknown latent concept space embedded in the LLM. We model a subset of the algorithms embedded in the LLM's latent concept space $\mathcal{H}$ that are invoked by ICL (Middle), to predict the LLM outputs $y$. With very little context (Left), GPT-3.5+ generates subjectively random sequences, whereas with enough context matching a simple formal language ($x = $ `001001001001` $= (001)^4$), it begins deterministically repeating patterns in $x$.

et al., 2022; Perez et al., 2022; Prystawski & Goodman, 2023; Turpin et al., 2023) and mechanistic neuron- and circuit-level understanding of pre-trained model capabilities on the other (Nanda et al., 2023; Goh et al., 2021; Geva et al., 2020; Belinkov, 2022; Li et al., 2022; Wang et al., 2022; Chughtai et al., 2023; Gurnee et al., 2023; Foote et al., 2023; Lubana et al., 2023; Lieberum et al., 2023; Barak et al., 2022; Liu et al., 2022b; Zhong et al., 2023) (also see App. C). Unlike related work of feature interpretability in vision neural networks, we analyze ICL and text generation dynamics over output sequences, and unlike mechanistic work studying ICL, we observe learning dynamics in black box input-output behavior of state-of-the-art LLMs.

Specifically, we focus on interpreting latent concepts in LLMs in the domain of random sequences of binary values (Fig. 1). Random binary sequences are a minimal domain that has been studied extensively in statistics, formal language theory, and algorithmic information theory (Sipser, 1996; Chaitin, 1966; Li et al., 1997). We use this domain to systematically test few-shot learning as a function of context length $|x|$ across different binary token sequences $x$. Moreover, language generation trajectories over binary sequences can also be analyzed and visualized more easily than typical user-chatbot interaction trajectories (Zhang et al., 2023a; Berglund et al., 2023) since the token-by-token branching factor is only two. Random binary sequences have also been a target domain in cognitive science (specifically, *subjective randomness*), where researchers have studied the mechanisms and concepts that underlie how people generate random binary sequences or evaluate the randomness of a sequence (Falk & Konold, 1997; Tversky & Gilovich, 1989; Griffiths et al., 2018).

Similar to computational cognitive theories of human concept learning (Chater & Vitányi, 2003; Dehaene et al., 2022; Piantadosi et al., 2012; Goodman et al., 2008; Ullman et al., 2012; Ullman & Tenenbaum, 2020), we argue that ICL can be interpreted as under-specified program induction, where there is no single "correct" answer; instead, an LLM should appropriately re-weight latent algorithms. The domain of random sequences reflects this framing, in contrast to other behavioral evaluation methodologies, in that there is no correct answer to a random number generation or judgment task (Fig. 3). If the correct behavior is to match a target random process, then the best way to respond to the prompt *Generate N flips from a fair coin* is a uniform distribution over the tokens `Heads` and `Tails`, instead of a specific ground truth token sequence. If instead the correct behavior is to match how humans generate random sequences, the LLM must represent a more complex algorithm that matches the algorithm used by people. Overall, we make the following contributions in this work.

- **Evaluating ICL as Bayesian model selection in a practical setting (Sec. 2).** We find evidence of in-context learning operating as Bayesian model selection in LLMs in the wild, compared to prior work that trains smaller toy neural networks from scratch.

- **Subjective Randomness as a domain for studying ICL dynamics (Sec. 3).** We introduce a new domain, *subjective randomness*, which is rich enough to be theoretically interesting, but simple enough to enable analysis of complex behavioral patterns.

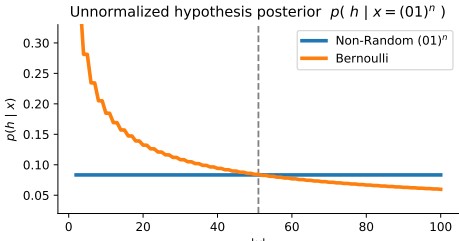 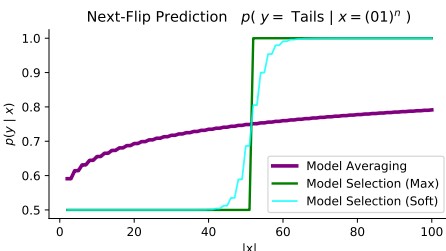

Figure 2: **Sharp changes in the predictive distribution $p(y|x)$ suggest model selection** (Left) Here we show an illustrative example of a simple Bayesian model with two hypotheses in $\mathcal{H}$: a random Bernoulli hypothesis, and a deterministic Non-Random hypothesis for the concept $(01)^n$. When hypothesis posteriors $p(h|x)$ cross as a function of additional data, i.e. larger $|x|$, the predictive distribution $p(y|x)$ for model selection suddenly transitions from one pattern of behavior to another (Right). With model averaging, instead there is a steady, gradual change in $p(y|x)$ (see App. D for further explanation). We find analogous S-shaped curves in ICL with LLM predictive distributions $p(y|x)$ in GPT-3.5+, suggesting model selection rather than model averaging (Fig. 4, 5). These phase changes are not predicted by theories of ICL as linear regression (Akyürek et al., 2022) or few-shot learning (Brown et al., 2020; Wei et al., 2022b), where loss steadily decreases with more context examples in $x$.

- **Sharp transitions in abilities to in-context learning dynamics (Fig. 4, 5).** We systematically analyze in-context learning dynamics in LLMs without observing or updating model weights, demonstrating sharp phase-changes in model behavior. This is a minimal, interpretable example of how the largest and most heavily fine-tuned LLMs can suddenly shift from one pattern of behavior to another during text generation, and supports theories of ICL as model selection.

## 2 IN-CONTEXT LEARNING AS MODEL SELECTION

Bayesian inference is a key methodological tool of cognitive modeling, and recent work has framed in-context learning as Bayesian inference over models (Xie et al., 2022; Li et al., 2023b; Hahn & Goyal, 2023). The output tokens $y$ an LLM produces given input tokens $x$ can be modeled as a posterior predictive distribution $p(y|x)$ that marginalizes over latent concepts $c$ in the LLM: $p(y|x) = \int_{c \in \mathcal{C}} p(y|c) \, p(c|x)$ In other words, a context $x$ will activate latent concepts $c$ in an LLM according to their posterior probability $p(c|x)$, which the model marginalizes over to produce the next token $y$ by sampling from the posterior predictive distribution. This inference process takes place in network activation dynamics, without changing model weights.

In our experiments, we assume a hypothesis space $\mathcal{H}$ that approximates the latent space of LLM concepts used when predicting the next token, i.e.,

$$p(y|x) = \sum_{h \in \mathcal{H}} p(y|h) \, p(h|x),$$

We assume that an LLM operates over next-token prediction only, and so each generated token $y_{t-1}$ will be appended to the context when the following token is sampled, i.e. $x_t = (x_{0,...,t-1}, y_{t-1})$. With this framework, we can understand ICL dynamics through the lens of hypothesis search. An LLM might not compute the sum in the above equation by taking a weighted average of latent capabilities. An LLM might perform Bayesian model selection, instead of model averaging, where for any given $x$, only a single latent concept $h$ approximates the posterior predictive distribution: $h^* = \text{argmax}_h \, p(h|x), \quad p(y|x) = p(y|h^*)$. We provide a visual demonstration of how Bayesian model selection leads to sharp changes in the predictive distribution $p(y|x)$ with a toy model in Fig. 2.

Cognitive scientists have used Bayesian predictive and posterior distributions to model learning dynamics in human adults and children, as well as in non-human animals (Tenenbaum, 1998; Goodman et al., 2008; Ullman & Tenenbaum, 2020). A discrete hypothesis space in a probabilistic model can give clear and meaningful explanations to learning patterns analogous to mode collapse and phase transitions in deep learning. When behavior suddenly shifts from one pattern, or mode, of behavior to another, this can be understood as one hypothesis coming to dominate the posterior $p(h|x)$ as $x$ grows in scale. This theory can explain conceptual shifts observed when children learn intuitive theories of domains such as intuitive physics (Ullman et al., 2012) and learning to count (Piantadosi et al., 2012). Such cognitive analysis of behavioral shifts as generated from a shift in

posterior probability between one hypothesis to another parallels recent work on learning dynamics in training and prompting LLMs. Sharp phase changes have been observed when training neural networks, corresponding to the formation of identifiable mechanisms such as modular addition circuits or induction heads (Nanda et al., 2023; Olsson et al., 2022; Zhong et al., 2023; Bai et al., 2023), and can be interpreted as Bayesian model selection (Hoogland et al.). Also see App. A, C.

To our knowledge, we provide the first demonstration of such sharp phase changes in ICL, particularly with state-of-the art blackbox LLMs. The minimal model used by Xie et al. (2022), as well as other few-shot learning work (Brown et al., 2020), show linearly or logarithmically decreasing loss curves in ICL, rather than S-shaped curves with stable plateaus as we find. Michaud et al. (2023) propose that neural network knowledge and skills may be quantized into discrete chunks, and so training might consist of many such phase changes which disappear into a smooth curve when test loss is averaged over many samples. We propose that In-Context Learning may also consist of sudden phase changes, as well as other complex learning dynamics, which may be explained as search over a discrete latent concept space. Such phase changes may occur in typical few-shot learning domains (Wei et al., 2022b; Brown et al., 2020), where smooth, continuously decreasing loss curves might hide complex ICL dynamics corresponding to shifts in the latent algorithms driving an LLMs behavior.

## 3 ALGORITHMIC AND SUBJECTIVE RANDOMNESS

Next, we describe the domain of randomness explored in this work, and provide relevant background from computational cognitive science. To build intuition, consider: which sequence of coin flips is more random, TTHTHHTH or HTTTTTTT? While the odds of a fair coin producing either sequence are the same (1 in 256), most people say that the first sequence "looks more random". In a bias termed *the Gambler's Fallacy*, people reliably perceive binary sequences with long streaks of one value as 'less random', and judge binary sequences with higher-than-chance alternation rates as 'more random' than truly random sequences (Falk & Konold, 1997). In addition to not using simple objective measures when judging the randomness of sequences, people also have a hard time generating sequences that are truly random (Oskarsson et al., 2009; Griffiths et al., 2018; Gronchi & Sloman, 2021; Meyniel et al., 2016; Planton et al., 2021).

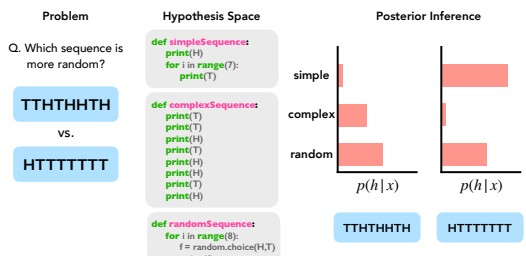

Figure 3: **Determining whether a sequence is random can be viewed as search for a simple program that could generate that sequence.** HTTTTTTT can be described with a short deterministic program `simpleSequence`, and has a higher $p(h)$ according to a simplicity prior, compared to TTHTHHTH and `complexSequence`. Both sequences can be generated by `randomSequence`, but with lower likelihood $p(x|h)$.

Cognitive scientists studying *Subjective Randomness* model how people perceive randomness, or generate data that is subjectively random but algorithmically non-random, and have related this problem to algorithmic information theoretic definitions of randomness. One way to study this is to ask people whether a given data sequence was more likely to be generated by a Random process or a Non-Random process (Fig. 3). HTTTTTTT might be described with a shorter program—*one H followed by seven Ts* – than HTTHTHHT, which cannot be compressed as well—*H followed by two Ts, then one H, then one T, then two Hs, then one T*. Viewing this task as program induction, a natural prior $p(h)$ over program hypotheses assigns higher weights to programs with shorter description lengths, or lower complexity. Similar to our description of model selection, or hypothesis search, compared to model averaging, the space of non-random programs can be simplified to a search for the single shortest program consistent with the data $x$. This search is equivalent to computing the Kolmogorov complexity of a sequence $K(x)$ (Li et al., 1997) and has motivated the use of "simplicity priors" in a number of domains in computational cognitive science Chater & Vitányi (2003); Goodman et al. (2008); Piantadosi et al. (2012).

As in Griffiths & Tenenbaum (2003); Griffiths et al. (2018), we define subjective randomness of a sequence as the ratio of likelihood of that sequence under a random versus non-random model,

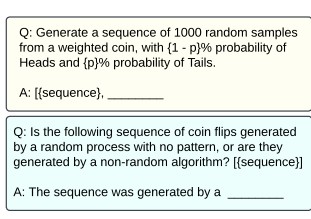

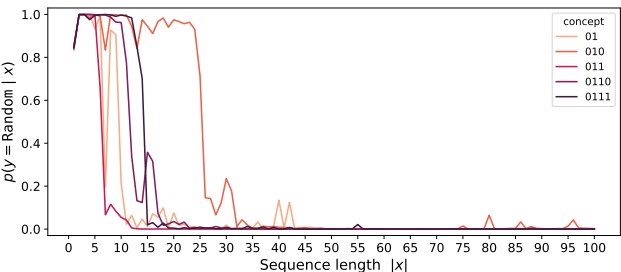

Figure 4: **Prompt templates and GPT-3.5 Randomness Judgments** (Left) Prompt templates used for the Randomness Generation (top) and Judgment (bottom) tasks. `{sequence}` is substituted with a list of context flips $x$, e.g. `Heads`, `Tails`, `Tails`, `Heads`. `{p}` is substituted with the probability of Tails, and `{1 − p}` with the probability of Heads. (Right) In Randomness Judgment tasks, we observe sharp transitions in the predictive distribution $p(y = \texttt{random}|x)$ for GPT-3.5, from high-confidence in $x$ being generated by a random process, to high confidence in a non-random algorithm, as additional data $x$ is provided from the same concept. See App. G for additional concepts.

i.e., $\texttt{randomness}(x) = \log P(x|\texttt{random}) - \log P(x|\texttt{non-random})$. The likelihood given a truly random Bernoulli process is $p(x|\texttt{random}) = 2^{-|x|}$, since each of $2^{|x|}$ binary sequences of length $|x|$ has equal probability. The non-random likelihood $p(x|\texttt{non-random}) = 2^{-K(x)}$ is the probability of the minimal description length program that generates $x$, equivalent to Bayesian model selection: $P(x|\texttt{non-random}) = \max_{h \in \mathcal{H}} p(x|h)\, p(h)$. See App. K for more background. In theory, the space $\mathcal{H}$ of non-random models encompasses every possible computation; in this work, we study a small subset of this space that includes formal languages and probabilistic models inspired by psychological models of human concept learning and subjective randomness (Griffiths et al., 2018; Nickerson, 2002; Hahn & Warren, 2009). Specifically, we consider Bernoulli processes, regular languages, Markov chains, and a simple memory-constrained probabilistic model as candidates for $\mathcal{H}$.

## 4 EXPERIMENTS

Our experiments aim to evaluate how LLMs operate in the domain of Subjective Randomness. Specifically, our goal is to understand whether ICL and text generation dynamics suggest model selection, or hypothesis search, over a discrete concept space. We first evaluate whether there is an emergent algorithm for generating subjectively random sequences, and if so, whether it is similar to cognitive models of human subjective randomness (Fig. 6, 7). We use open-ended text generation experiments (Fig. 4, Left) and analyze binary sequences produced by LLMs, and compare LLM sequences to a Bernoulli distribution and our cognitively inspired (Falk & Konold, 1997; Hahn & Warren, 2009) Window Average model. Second, we evaluate whether ICL dynamics in searching over random and non-random hypotheses demonstrate ICL operating as model averaging, where learning is smooth and continuous, or as model selection, or hypothesis search, where sudden, dramatic changes can occur during learning.

**Randomness Generation and Judgment Tasks** In order to assess text generation dynamics and in-context concept learning, we evaluate LLMs on random sequence **Generation** tasks, analyzing responses according to simple interpretable models of *Subjective Randomness* and *Formal Language Learning*. In these tasks, the model generates a sequence $y$ of binary values, or *flips*, comma-separated sequences of `Heads` or `Tails` tokens. We also analyze a smaller set of randomness **Judgment** tasks, with the prompt: *Is the following sequence of coin flips generated by a random process with no pattern, or are they generated by a non-random algorithm?*. In both cases, $y$ is a distribution over tokens with two possible values. Outputs $y$ consist of a single `Random` or `Non` token, indicating whether the sequence was generated by a random process with no correlation, or some non-random algorithm. We analyze dynamics in LLM-generated sequences $y$ simulating a weighted coin with specified $p(\texttt{Tails})$, with $|x| \approx 0$. An initial flip '`Heads`' is used so the model's flips will be formatted consistently. Prompt context in these experiments includes a specific probability, shown in Fig. 4, where the last __ marks where the model begins generating tokens $y$. We collect 200 output sequences $y$ for each LLM at each $P(Tails) \in [.05, .1, .2, .3, .4, .49, .5, .51, .60, .70, .80, .90, .95]$, cropping output tokens to $|y| = 50$ to limit cost and for simplicity. Temperature parameters are set to 1, and other parameters follow OpenAI API defaults; see App. B for more details.

In the Judgment task, $x$ includes the prompt question and the full sequence of flips. We systematically vary prompt context $x$ by varying the number of flips, denoted $|x|$. We test few-shot learning in LLMs

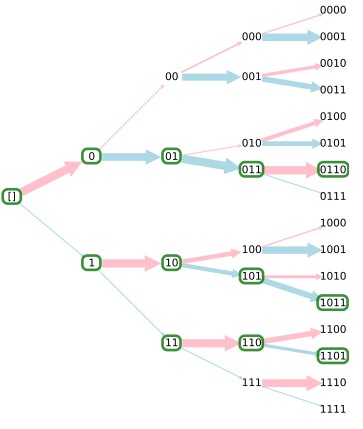 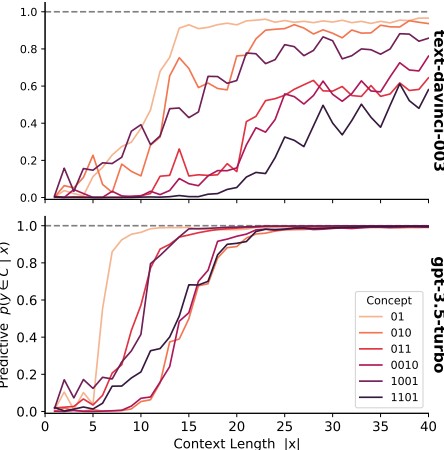

Figure 5: **With enough context, GPT-3.5 learns simple formal language concepts, transitioning from generating subjectively random sequences to only generating values matching the concept** (Left) GPT-3.5 next-token predictive distribution $p(y|x)$ visualized as a binary tree, where red arrows correspond to *Heads* (0), blue arrows to *Tails* (1), and nodes matching the target concept $C = (011)^n$ are dark green. $p(y|x)$ changes with varying $|x|$, here $|x| = 39$ and next-token predictions strongly follow the concept $C$. Also see Fig 15, 16 in Appendix. (Right) In-context learning dynamics for simple formal languages $x = (\texttt{HTH})^n$ (010) and $x = (\texttt{HTT})^n$ (011) as a function of context length $n = |x|$. Prediction accuracy computed as the total probability mass of $p(y|x)$ assigned to valid continuations of the formal language $x$, as a function of prediction depth $d = |y|$ and context length $|x|$. Curves shown are for depth $d = 6$, where only 3 out of 64 possible length-6 binary sequences $y$ match concept $C$. Note: bottom model is $\texttt{gpt-3.5-turbo-instruct-0914}$. Also see App. F.

by evaluating output behavior on specific bit sequences, for example $(\texttt{01})^n$ with varying $n$ (e.g. "$\texttt{Heads, Tails, Heads, Tails, }\ldots$"), as well as zero-shot language generation dynamics when $x$ is empty and $|x| = 0$ (in practice, we initialize $x$ with a single flip to help the LLM match the correct format). Unlike typical cases of few-shot learning, where high-variance samples ('shots') are drawn from a broad category such as "chain-of-thought solutions" (Wei et al., 2022b; Brown et al., 2020), all data in a sequence $x$ corresponds to a single unobserved program.

**Subjective Randomness Models**  We compare LLM-generated sequences to a ground truth "random" Bernoulli distribution with the same mean ($\mu = \overline{y}_{\text{LLM}}$), to a simple memory-constrained probabilistic model, and to Markov chains fit to model-generated data $y$. The Gambler's Fallacy is the human bias to avoid long runs of a single value (*'Heads'* or *'Tails'*) consecutively in a row (Falk & Konold, 1997; Nickerson, 2002). Hahn & Warren (2009) theorize that the Gambler's Fallacy emerges as a consequence of human memory limitations, where 'seeming biases reflect the subjective experience of a finite data stream for an agent with a limited short-term memory capacity'. We formalize this as a simple *Window Average* model, which tends towards a specific probability $p$ as a function of the last $w$ flips: $p(y|x) = \max(0, \min(1, 2p - \overline{x}_{t-w,\ldots,t}))$.

**Sub-Sequence Memorization and Complexity Metrics**  Bender et al. (2021) raise the question of whether LLMs are 'stochastic parrots' that simply copy data from the training set. We test the extent to which LLMs generate subjectively random sequences by arranging memorized patterns of sub-sequences, rather than according to a more complex algorithm as our Window Average model represents. To measure memorization, we look at the distribution of unique sub-sequences in $y$. If an LLM is repeating common patterns across outputs, potentially memorized from the training data, this should be apparent in the distribution over length K sub-sequences. Since there are deep theoretical connections between complexity and randomness (Chaitin, 1977; 1990), we also consider the complexity of GPT-produced sequences. Compression is a metric of information content, and thus of redundancy over irreducible complexity (MacKay, 2003; Delétang et al., 2023), and neural language models have been shown to prefer generating low complexity sequences (Goldblum et al., 2023). As approximations of sequence complexity, we evaluate the distribution of Gzip-compressed file sizes (Jiang et al., 2022) and inter-sequence Levenshtein distances (Levenshtein et al., 1966).

**Formal Language Learning Metrics**  In our Formal Language Learning analysis, $x$ is a subset of regular expression repetitions of short token sequences such as $x \in (\texttt{011})^n$, where longer sequences $x$ correspond to larger $n$. This enables us to systematically investigate in-context learning of formal languages, as $|x|$ corresponds to the amount of data for inducing the correct program (e.g. $(\texttt{011})^n$)

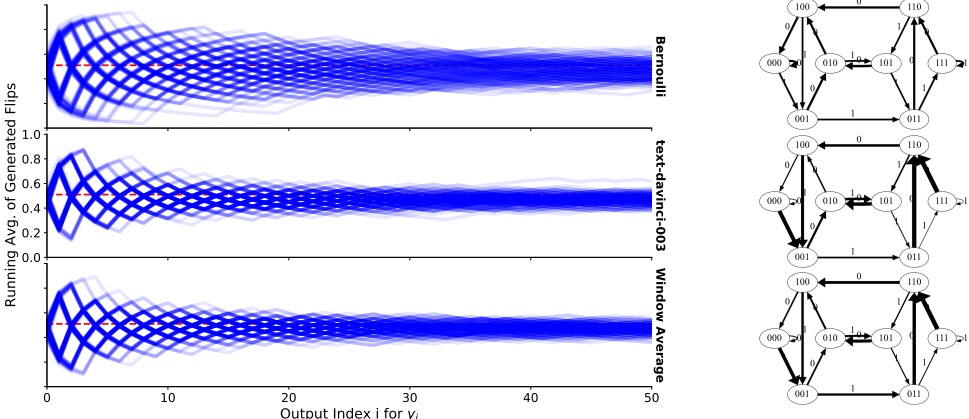

Figure 6: **GPT-3.5 generates subjectively random binary sequences that deviate from a Bernoulli process.** (Left) Running averages for flip sequences generated by each model, where 0 denotes 'Heads' and 1 denotes 'Tails'. Compared to a Bernoulli process (top), sequences generating using GPT (middle) and those of our Window Average model (bottom) stay closer to the mean, repeating the same patterns more often. (Right) Empirical conditional probabilities for a third-order Markov Chain fit to sequences $y$ generated by GPT-3.5 text-davinci-003, a Bernoulli process centered at the mean of GPT sequence $\overline{y}$, and our Window Average model ($w = 5$). In the simulated Bernoulli process, edges are fairly uniform; edges for GPT-3.5 and the Window Average model are non-uniform, but similar to one another.

from Bernoulli processes, as well as the space of all possible algorithms. A Markov chain of sufficiently high order $k$ can implicitly represent a regular language in its weights (see Fig. 6, Right). However, the number of parameters required is exponential in Markov order $k$, and weights are difficult to interpret. For example, a 3-order Markov chain fit to data $x = (011)^n$ will represent $(011)^n$ corresponds to high weights in the conditional probability distribution for $p(0|11)$, $p(1|01)$, $p(1|10)$, and low values for other conditional probabilities. To match the goal of the Generation task, we use a variation of the prompt in Fig. 4: *'Generate a sequence of 1000 samples that may be from a fair coin with no correlation, or from some non-random algorithm'*

In Randomness Judgment tasks, we assess formal concept learning dynamics by changes in the predictive distribution $p(y = \texttt{random}|x = C^{|x|})$, i.e. the probability of the final token being *'Random'* vs. *'Non'*, as a function of context length $|x|$. The method for measuring concept learning dynamics in Randomness Generation tasks requires computing $p(y|x)$ over possible binary flip sequences $y$, instead of $y$ being a single token as with Judgment tasks. In binary sequence Generation tasks, we estimate the predictive probability $p(y_t \in C|x, y_{0...t-1})$ assigned to a given regular language concept $C$ by computing the total probability mass of $p(y|x)$ for all unique token sequences $y_{0...d}$ up to some depth $d$, such that the $y$ exactly matches a permutation of the concept. If tokens in $x$ and $y$ have two possible values, 0 or 1, the probability table for $p(y_t|y_{0,...,t-1})$ is a weighted binary tree (see Fig. 5, Left) with depth $d$, where edges represent the next-token probability $p(y_t|y_{t-1})$, and paths represent the probability of a sequence of tokens. For example, with $C = (011)^n$, there will be 3 paths $y_{0...d}$ corresponding to the permutations $(011)^n, (101)^n, (110)^n$, out of $2^d$ possible sequences $y$, and so: $p(y \in (011)^n \mid x) = p(y = \text{'}0, 1, 1, 0, 1, 1, \ldots\text{'} \mid x) + p(y = \text{'}1, 1, 0, 1, 1, 0, \ldots\text{'} \mid x) + p(y = \text{'}1, 0, 1, 1, 0, 1, \ldots\text{'} \mid x)$. We use the notation $p(y \in C \mid x)$ in Fig. 5 (Right) to refer to this predictive probability of $y$ matching a concept $C$. In both Judgment and Generation tasks for formal concept learning, we use next-token prediction probabilities $p(y_t|y_{0,...,t-1}, x)$ directly from the OpenAI API, for cost efficiency and since this is equivalent to repeated token samples.

## 5 RESULTS

### 5.1 SUBJECTIVELY RANDOM SEQUENCE GENERATION

In the GPT3.5 and higher models—InstructGPT (text-davinci-003), ChatGPT (gpt-3.5-turbo) and GPT-4—we find an emergent behavior of generating seemingly random binary sequences (Fig. 6). *This behavior is controllable*, where different $p(\texttt{Tails})$ values lead to different means of generated sequences $\overline{y}$. However, the distribution of sequence means, as well as the distribution of the length of the longest runs for each sequence, deviate significantly from a Bernoulli distribution centered at $\overline{y}$, analogous to the Gambler's Fallacy bias in humans (see Fig. 7).

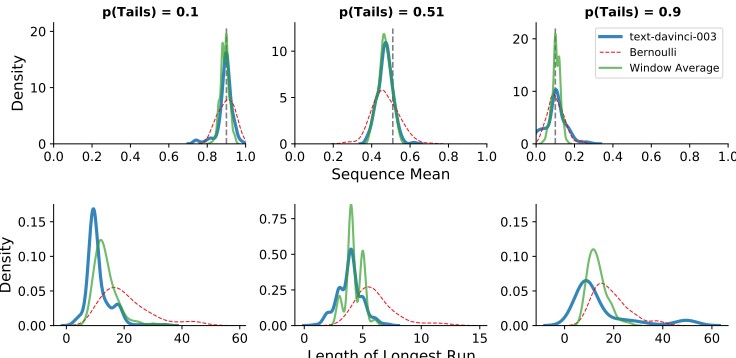

Figure 7: **GPT-3.5 shows a Gambler's fallacy bias of avoiding long runs.** (Top) Distribution of mean values of flip sequences ($\mu = \frac{1}{T} \sum_t y_t$) generated by GPT-3.5 (`text-davinci-003`) with the specified $p(\texttt{Tails})$, compared with a Bernoulli process and our Window Average model with the same mean as the GPT-3.5 flips. Flips generated by GPT approximately follow the expected mean $p(\texttt{Tails})$, but have lower variance than a Bernoulli distribution. (Bottom) Length of the longest run for each sequence, where a run is a sub-sequence of the same value repeating. In this case, we see a clear bias in GPT-3.5 to avoid long runs, with a similar pattern across all values of $p(\texttt{Tails})$ despite the x-axis changing in scale.

*Our Window Average model with a window size of $w = 5$ partly explains both biases*, matching GPT-generated sequences more closely than a Bernoulli distribution. This result demonstrates that ICL enables selection of *latent concepts in an LLM that operate as algorithms acting on data*. Our next experiment, on sub-sequence memorization, aims to further characterize this algorithm and address the alternative hypothesis that this behavior is generated by memorizing pre-training data.

For brevity, in the following sections we primarily analyze the behavior of `text-davinci-003`, referring to it simply as 'GPT-3.5' and including subsequent OpenAI GPT models ('gpt-3.5-turbo', 'gpt-4') as 'GPT-3.5+'. In App. L we show similar results with open-source LLMs. Our cross-LLM analysis shows that `text-davinci-003` is controllable with $p(\texttt{Tails})$, with a bias towards $\overline{y} = .50$ and higher variance in sequence means (though lower variance than a Bernoulli process). Both versions of ChatGPT we analyze (versions `gpt-3.5-turbo-0301` and `0613`) demonstrate similar behavior for $p(\texttt{Tails}) < .5$, but behave erratically with higher $P(\texttt{Tails})$ and the majority of sequences $y$ converge to repeating 'Tails'. Both versions of GPT-4 show stable, controllable subjective randomness behavior, where sequence means $\overline{y}$ closely follow the specified $p(\texttt{Tails})$, but have even lower variances than sequences generated by `text-davinci-003`. Earlier models do not show subjective randomness behavior, with `text-davinci-002` and `text-davinci-001` being heavily biased and uncontrollable, and `text-curie-001` generates sequences with $\overline{y} = .50$ regardless of $p(\texttt{Tails})$. Also see App. H.

## 5.2 SUB-SEQUENCE MEMORIZATION AND COMPLEXITY

We find significant differences between the distributions of sub-sequences for GPT-3.5 -generated sequences and sequences sampled from a Bernoulli distribution(see Fig. 8). This difference is partly accounted for with a Window Average model with a window size $w = 5$, although GPT repeats certain longer sub-sequences, for example length-20 sub-sequences, that are far longer than 5. However, the majority of sub-sequences have very low frequency, and though further experiments would be required to conclude that all sub-sequences are not memorized from training data, it seems unlikely that these were in the training set, since we find thousands of unique length-k (with varying k) sub-sequences generated at various values of $P(\texttt{Tails})$. *This indicates that GPT-3.5 combines dynamic, subjectively random sequence generation with distribution-matched memorization.* This suggests that ICL is indeed eliciting complex latent algorithms in subjective randomness tasks, and that this behavior cannot be easily attributed to memorizing sequences from the training distribution.

Note that with a specification of $p(\texttt{Tails}) = 50\%$, but not $49\%$, $51\%$ or other values, sequences $y$ generated by GPT-3.5+ are dominated by repeating 'Heads, Tails, Heads, Tails, . . . '. This pattern is consistent across variations of the prompts listed in Fig. 4, including specifying 'fair' or 'unweighted' instead of a 'weighted coin', and produces a visible kink in many cross-$p(\texttt{Tails})$ metrics (Fig. 25, 28, 29). Due to this, in Fig. 7 we show results for $p(\texttt{Tails}) = 51\%$. Across three metrics of sequence complexity—number unique sub-sequences, Gzip file size, and inter-sequence Levenshtein distance (see Fig. 29, 28 in Appendix)—we find that *GPT-3.5+ models, with the exception*

*of ChatGPT, generate low complexity sequences*, showing that structure is repeated across sequences and supporting Goldblum et al. (2023); Delétang et al. (2023). For mean Levenshtein distance and number of unique sub-sequences, ChatGPT generates higher complexity sequences than chance. We speculate that this phenomenon might be explained in future work by a simple cognitive model: if the LLM makes a sequence more subjectively random by avoiding repeating sub-sequences, like a human might, then it will produce sequences with higher-than-chance complexity overall.

### 5.3 DISTINGUISHING FORMAL LANGUAGES FROM RANDOMNESS

*GPT-3.5 sharply transitions between behavioral patterns, from generating subjectively random values to generating non-random sequences that perfectly match the formal language* (Fig. 5). We observe a consistent pattern of formal language learning in `gpt-3.5-turbo-instruct-0914` generating random sequences where predictions $p(y|x)$ of depth $d \geq 4$ are initially random with small $|x|$, and have low $p(y \in C|x)$ where $C$ is a given concept. This follows whether the prompt describes the process as samples from "a weighted coin" or "a non-random-algorithm". ICL curves for longer concepts are smoother for `text-davinci-003` and do not converge to $p(y|x) \approx 1$ as they do for `gpt-3.5-turbo-instruct-0914`. We do not explore the reason for this distinction or when this breakdown occurs; however, in App. G we analyze longer concepts in Randomness Judgment experiments, which are more cost-efficient.

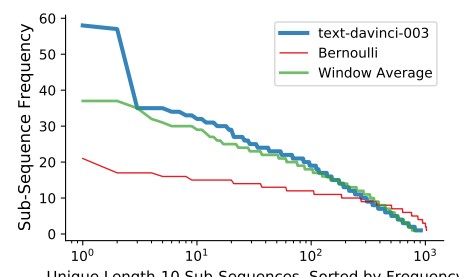

Figure 8: **GPT repeats certain sub-sequences more often than chance.** We find that GPT-3.5 repeats certain sub-sequences more often than is predicted by our Window Average model. While the Window Average model generates fewer unique sub-sequences than a Bernoulli process, this does not account for the bias in GPT-3.5 (`text-davinci-003`) to repeat many of the same sub-sequences. This disparity increases with longer sub-sequences (Fig. 26).

We also find *sharp phase changes in GPT-3.5 behavioral patterns in Randomness Judgment tasks across 9 binary concepts* (Fig. 4). These follow a stable pattern of being highly confident in that the sequence is Random (high $p(y = \text{random}|x)$ when $x$ is low, up to some threshold of context at which point it rapidly transitions to being highly confident in the process being non-random. Transition points vary between concepts, but the pattern is similar across concepts (also see Fig. 17, 18 in Appendix). These transitions support our theory of ICL as model selection, or discrete hypothesis search, rather than model averaging, as in Fig. 2.

## 6 DISCUSSION

We find strong patterns of random number Generation and Judgment in LLMs, including an emergent ability to generate low-variance, 'subjectively' random binary sequences, that can be partly explained with a memory-constrained moving window model. This behavior can also partly be explained as simple memorization and compression, but we observe algorithmic behavior that seems to go beyond mere parroting. We also find striking behavioral phase changes when GPT-3.5 learns to distinguish simple formal language concepts from randomness. This supports our framework of understanding ICL as Bayesian model selection and as under-specified program induction, and demonstrates how LLM behavioral patterns can change non-linearly with ICL as a function of context data. Our work is an initial step in a direction which may prove to be highly impactful: if new capabilities can suddenly, rather than gradually, emerge after a single additional data point is presented in-context, this presents major challenges to AI safety and LLM interpretability. LLM capabilities may suddenly emerge or disappear with new prompts, or with more 'shots', and LLM behavior may suddenly and dramatically shift over the course of a single user interaction. These phase changes may be common in ICL, unobserved by research in few-shot learning that averages over many samples when considering in-context learning curves, instead of considering individual learning trajectories as we do. Future work may characterize concept learning dynamics underlying other LLM capabilities, including applications in usability and safety (App. C). If ICL is analogous to hypothesis search in human cognition, a compelling research direction would be to further understand the defining features and limits of in-context learning dynamics using theories and tools from computational cognitive science.

## REPRODUCIBILITY STATEMENT

All code and data used for this paper are available at https://github.com/ebigelow/ICL-Random-Binary. In Appendix L we replicated our results across a selection of open-source LLMs available through the HuggingFace Transformers API (Llama 2, Mixtral, Tulu 2). We found similar results to GPT-3.5+ on Randomness Generation tasks varying $p(Tails)$ with LLMs with 50+ billion parameters (Fig. 32 & 33) and similar results on Randomness Judgment with tulu-2-dpo-70b (Fig. 30). As of January 2024, the primary LLM analyzed in this paper text-davinci-003 is no longer publicly accessible.

## ACKNOWLEDGEMENTS

ESL's time at University of Michigan was partially supported via NSF under award CNS-2008151.

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

# Appendices

## A RELATED WORK

**Formal languages and transformers.** A number of recent works explore how transformers and other neural language models learng formal languages (Delétang et al., 2022; Weiss et al., 2021; Shi et al., 2022; Allen-Zhu & Li, 2023; Bhattamishra et al., 2020; Wen et al., 2023; Liu et al., 2023; 2022a; Merrill & Sabharwal, 2023; Merrill et al., 2023; Merrill & Sabharwal, 2022). One common theme is that neural networks often learn 'shortcuts', degenerate representations of formal languages that fail out-of-samples.

**In-Context Learning as Bayesian inference** A number of recent works frame ICL as Bayesian model selection (Xie et al., 2022; Li et al., 2023b; Akyürek et al., 2022; Hahn & Goyal, 2023; Bai et al., 2023). Two key differences in our work are: first, we analyze state-of-the-art LLMs based on behaviors alone, whereas prior work trains models from scratch on synthetic data and analyzes model parameters directly. Second, we consider Bayesian inference as an empirical modeling framework, as well as a theory, whereas these works only do the latter.

**Mechanistic interpretability of transformer models** Prior work has characterized specific circuit-level implementations of simple high-level behaviors such as sequence copying, modular addition, and other primitive computational operations (Nanda et al., 2023; Goh et al., 2021; Geva et al., 2020; Belinkov, 2022; Li et al., 2022; Wang et al., 2022; Chughtai et al., 2023; Gurnee et al., 2023; Foote et al., 2023; Lubana et al., 2023; Lieberum et al., 2023; Barak et al., 2022; Liu et al., 2022b; Zhong et al., 2023). Our work differs in that we model hypothetical algorithms to characterize LM output behavioral patterns, without observing underlying activation patterns. We see this as analogous to cognitive science complementing neuroscience in the understanding of human cognition. We characterize the high-level "cognitive" representations in LLMs as a step towards connecting low-level explanations of neural circuits, such as induction heads, with sophisticated high-level behaviors that are characteristic of LLMs.

**Language model evaluations** Our work resembles evaluation benchmarks such as BIG-Bench (Srivastava et al., 2022) that use behavior alone to evaluate LM understanding and reasoning. However, as described in the main text, the domain of subjective randomness is fundamentally different in that there is no "correct" answer. Linguistic probing attempts to characterize the structure of LM representations, but unlike our work, is a function of hidden unit activations rather than output behavior.

**LLM Text Generation Dynamics** Work on chain-of-thought reasoning in LLMs demonstrates how a few exemplars of detailed solutions or even a simple prompt like "let's think this through step-by-step" can dramatically impact model performance (Wei et al., 2022b; Kojima et al., 2022; Srivastava et al., 2022), but typically only the model's final answer is analyzed, not the trajectory of its intermediate steps. Our memory-constrained Window Average model, inspired by Hahn & Warren (2009), is similar in spirit to the claim of Prystawski & Goodman (2023), that '[chain-of-thought] reasoning emerges from the locality of experience'. Zhang et al. (2023a) demonstrate that invalid reasoning can snowball in LLMs, where hallucinations during intermediate steps lead to hallucinations in the final answer.

**Random number generation in LLMs** Renda et al. (2023) explore random number generation in LLMs, in addition to cursory explorations by (janus, 2022; Karpathy, 2023). These investigations do not analyze dynamics of sequence generation, nor do they ground their analysis, as we do, in theories of ICL as Bayesian model selection and the cognitive science of subjective randomness. Ortega et al. (2019) uses a similar domain as ours with random binary sequences and has a similar binary tree visualization over possible sequences, but they train models from scratch and analyze model hidden states, rather than behavioral trajectories as we do.

**Bayesian program learning in cognitive science** Our work is inspired by computational cognitive science work that theoretically treats concepts as programs, and empirically uses structured Bayesian models to understand human cognition in various domains (Tenenbaum, 1998; Tenenbaum et al., 2011; Ullman et al., 2012; Ullman & Tenenbaum, 2020). We use models based on the cognitive science of subjective randomness (Falk & Konold, 1997), drawing particularly on the Bayesian

program induction definitions of subjective randomness in Griffiths & Tenenbaum (2003; 2004); Griffiths et al. (2018). Our method of studying learning as probabilistic inference over formal languages with varying $|x|$ is also similar to Goodman et al. (2008); Piantadosi et al. (2012); Yang & Piantadosi (2022); Bigelow & Piantadosi (2016), who use more sophisticated grammar-based models of concept learning.

## B  ADDITIONAL EXPERIMENTAL DETAILS

All calls were made with the OpenAI API, using default parameters including — important to our analysis — a temperature parameter of 1.0.

Since ChatGPT (not including `gpt-3.5-turbo-instruct`) and GPT-4 use a ChatCompletions API instead of Completions, we re-formatted the prompts in Fig. 4 to follow user/assistant/system prompt format. The following prompts were used:

| System | Your responses will only consist of comma-separated "Heads" and "Tails" samples.
Do not repeat the user's messages in your responses. |
| --- | --- |
| User | Generate a sequence of 1000 random samples from a weighted coin, with {1 - p}% probability of Heads and { p }% probability of Tails. |
| Assistant | [ { sequence } |

Although results are not shown here, for Randomness Judgment experiments, we also tested `text-davinci-003` with prompts other than the one in Fig. 4, including specifying *a non-random algorithm* instead of *a weighted coin, with { 1 - p }% probability ...*, and found similar results of concept learning dynamics as in Fig. 4, 18.

We chose the format *"Heads", "Tails", ...* specifically to avoid the alternate interpretation you describe, of some patterns merely being tokenization artifacts (Fig. 9), since flips with shorter string formats, such as 0s and 1s with no comma separating them, often get chunked into different tokens (*"01", "0000"*, etc.). We specify '1000 random samples' in our Generation prompt to avoid issues with GPT generating too few tokens, where for example if the *'50 samples'* is specified, less than 50 flips will be generated in some cases.

In our analyses, we omit the probabilities of comma tokens $p(y = \text{`,'} \mid x)$, as well as partial flip tokens - in particular $p(y = \text{`ails'} \mid x = \text{` T'})$ - since these are $approx 1$ and nearly deterministic. When analyzing GPT outputs in our flip Generation analyses, we segment flips tasks according to tokens that begin with *'h'* or *'t'*, after dropping spaces and converting to lower-case.

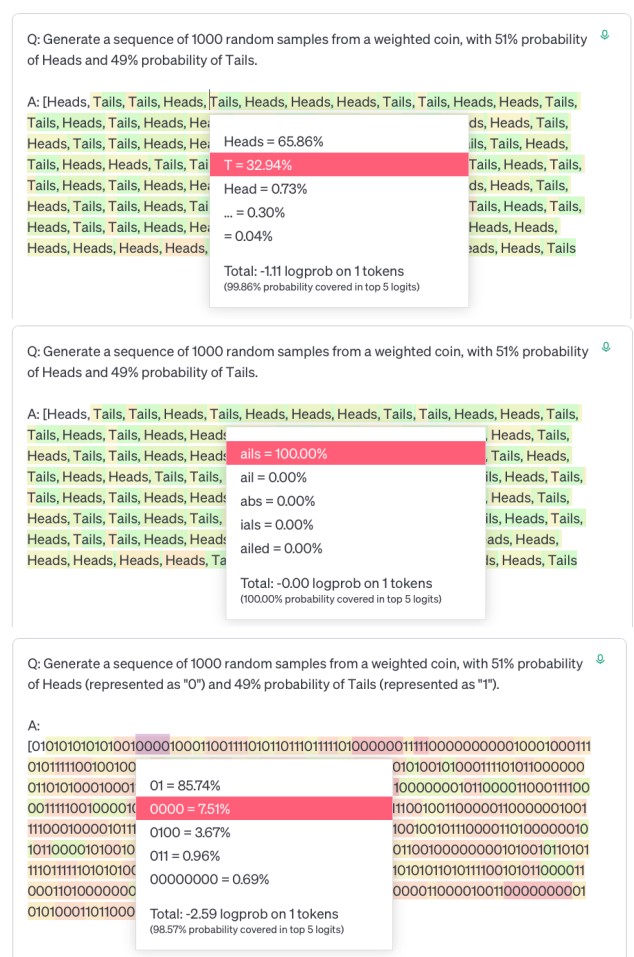

Figure 9: **Consistent tokenization with 'Heads, Tails, ...' compared to '010 ...'** With comma-separated lists of *'Heads'* and *'Tails'* flips, tokens consist of *'Heads'*, *'T'*, *'ails'*, *','*, as well as occasional erroneous tokens such as *'Head'*. With comma-less *'0'* and *'1'* tokens, the GPT tokenizer chunks many binary sub-sequences into individual tokens.

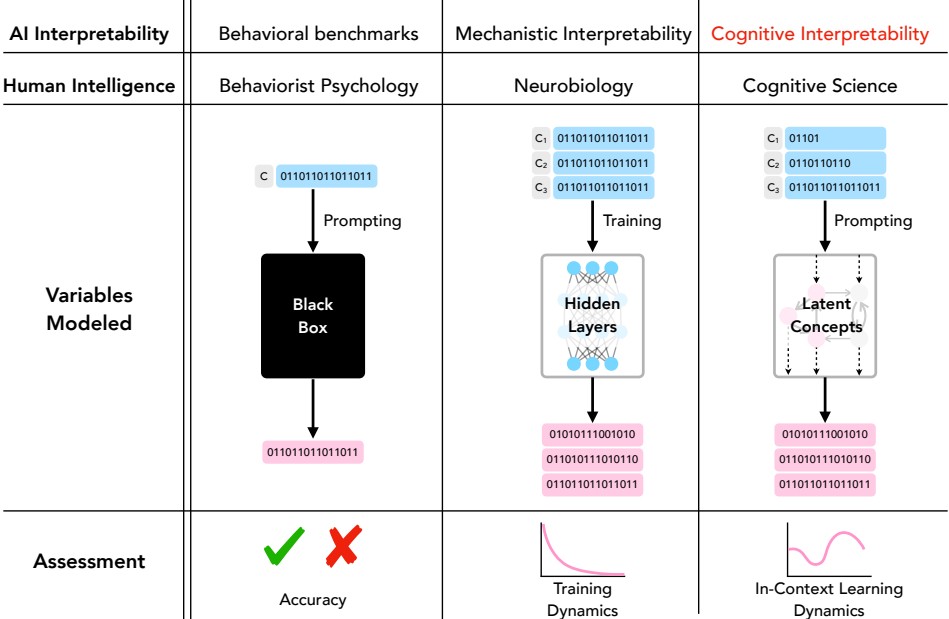

Figure 10: **Cognitive Interpretability in the context of prior work** Cognitive Interpretability studies in-context learning dynamics in LLMs, positing theories of the latent concepts enabling behavioral capabilities. It is a middle ground between behavioral benchmarks, which treat models as black boxes and evaluate hit-or-miss accuracy, and mechanistic interpretability, which studies toy models and training loss dynamics, analogous to how cognitive science is a middle ground between behaviorist psychology and neurobiology.

## C    COGNITIVE INTERPRETABILITY

*Cognitive Interpretability* (Fig. 10) is a middle ground between shallow test-set evaluation benchmarks on one hand (Srivastava et al., 2022; Saparov & He, 2022; Min et al., 2022; Lanham et al., 2023; Ganguli et al., 2023; Bowman et al., 2022; Kadavath et al., 2022; Perez et al., 2022; Prystawski & Goodman, 2023; Turpin et al., 2023) and mechanistic neuron- and circuit-level understanding of pre-trained model capabilities on the other (Nanda et al., 2023; Goh et al., 2021; Geva et al., 2020; Belinkov, 2022; Li et al., 2022; Wang et al., 2022; Chughtai et al., 2023; Gurnee et al., 2023; Foote et al., 2023; Lubana et al., 2023; Lieberum et al., 2023; Barak et al., 2022; Liu et al., 2022b; Zhong et al., 2023) (also see App. C). Unlike related work of feature interpretability in vision neural networks, we analyze ICL and text generation dynamics over different input data $x$ and outputs $y$, and unlike mechanistic work studying ICL, we observe learning dynamics in black box input-output behavior of state-of-the-art LLMs.

This is inspired by cognitive science work where learning dynamics are observed in humans, and these learning dynamics are explained with computational theories and probabilistic programming models. For example, children 3.5–5 years old learning to count undergo a dramatic conceptual shift from knowing the meanings of only a few number words ("one", "two") to a full inductive understanding of counting, which can be modeled as Bayesian model selection with a simplicity prior over models (Piantadosi et al., 2012). Another example is children learning intuitive theories of physics and social reasoning, where theories may spontaneously shift as 'a new concept is acquired'. This can be explained, similar to our models, as a shift in the posterior space, which influences hypothesis search (Ullman et al., 2012). The work of Ullman et al. (2012) is also of note, since they make a similar point to the neural network Quantization hypothesis Michaud et al. (2023) – that sharp discontinuities ('phase changes') are common in learning, but averaging over samples and individuals turns these discontinuities into a smooth learning curve – but in the context of children's intuitive theory learning and hypothesis search over probabilistic programs. Similar theories are also proposed about theory change in adults, for example, in scientific concepts dramatically shifting as paradigms change (Carey, 2000; Nersessian, 2010).

In-Context Learning and emergent capabilities in LLMs present unique and novel challenges and opportunities for AI interpretability, as well as for cognitive science. Computational cognitive science provides rich theory and myriad domains which can be used for evaluating LLM learning

and behavioral dynamics. Our approach is distinct from approaches such as Binz & Schulz (2023) which evaluate LLMs on cognitive psychology benchmarks, since such prior work examines single-token probabilities but does not consider, as we do, in-context learning dynamics or applications of computational theories of cognition (Tenenbaum, 1998; Tenenbaum et al., 2011; Ullman & Tenenbaum, 2020; Piantadosi, 2021) to understanding LLM capabilities. Our approach is also distinct from using LLMs as models of human cognition. Though we use theories and models from cognitive science to understand LLMs, we do not assume that what the LLM does should be human-like.

While we propose a new term, Cognitive Interpretability, we opt to provide a loose definition rather than a strict one. The purpose of this term is to emphasize a missing piece of the LLM interpretability puzzle, which is not given as much attention as evaluation benchmarks or mechanistic circuit-level understanding. Mechanistic interpretability is sometimes described as *the neuroscience of deep learning*. We suggest a role for *the cognitive science of deep learning*, distinct from creating new evaluation benchmarks, which seeks to understand the high-level algorithms underlying complex patterns of behavior and learning.

## D   BAYESIAN MODEL SELECTION EXAMPLE

This section further explains the illustrative example presented in Fig. 2.

This example compares two models: a ("Random") Bernoulli process with $p = .5$, and a ("Non-Random") deterministic concept $(01)^n$. The likelihood of the Random hypothesis is $p(x|h) = (.5)^{|x|}$, which decreases as a function of $|x|$. The likelihood of the Non-Random hypothesis is a fixed constant, for simplicity, $p(x|h) = 1$.

The hypothesis posteriors shown in Figures 2, 11 are a product of these likelihood functions with a hypothesis prior $p(h)$, following Bayes' theorem: $p(h|x) \propto p(h) \, p(x|h)$. In simple terms, the prior $p(h)$ determines the relative y-position of the two likelihood curves, which determines the precise point along $|x|$ at which the curves intersect. We arbitrarily chose priors $p(h_{\text{Non}}) = .25$ and $p(h_{\text{Non}}) = .75$.

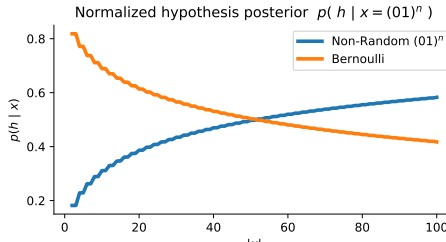 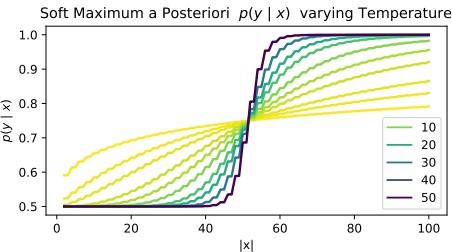

Figure 11: (Left) Normalized hypothesis posteriors from Fig. 2 Left. These are normalized so that probability of all hypotheses sums to one $\sum_h p(h|x) = 1$, i.e. $p(h = \text{Bernoulli}|x) + p(h = \text{Non-Random}|x) = 1$. (Right) The soft model selection curve shown in Fig. 2 Right, with varying temperature parameter $\tau$ interpolating $p_\tau(y|x)$ between Model Averaging ($\tau = 1$, yellow line) and Model Selection ($\tau \to \infty$, dark blue line).

In the right side of Figure 2, we compare Model Averaging $p(y|x) = \sum p(y|h) \, p(h|x)$ to Model Selection $h^* = \text{argmax}_h \, p(h|x), \; p(y|x) = p(y|h^*)$. With model selection, at the point when the the the two hypothesis posteriors cross, the predictive distribution $p(y|x)$ suddenly shifts as the non-random hypothesis becomes selected for $h^*$ instead of the random one. This example illustrates how model selection, or maximum likelihood hypothesis search, can explain sharp phase changes in model behavior according to the predictive distribution $p(y|x)$.

Since the phase change in-context learning curves we observe are smooth, as well as those observed during neural network training and mechanism formation, we also show a 'soft' model selection curve. The soft model selection curve interpolates between model averaging and model selection (see Fig. 11, Right):

$$p_\tau(y|x) = \sum_h p(y|x) \Big( \frac{p(h|x)^\tau}{\sum_{h'} p(h'|x)^\tau} \Big)$$

where $\tau$ is a temperature parameter such that when $\tau = 1$, the function $p_\tau(y|x)$ is equal to the model averaging definition above, and when $\tau$ grows arbitrarily large, $p_\tau(y|x)$ approaches the model selection definition above.

Temperature $\tau$ in this model is distinct from temperature in LLMs, which leads models to greedy decode as $\tau \to 0$, i.e. $p(y|x) = \max_h p(y|h) \, p(h|x)$. This maximizes over the posterior predictive distribution, which includes both $p(y|h)$ and $p(h|x)$, whereas in our example, $h^*$ only maximizes over the hypothesis posterior $p(h|x)$.

## E  FORMAL CONCEPT LEARNING IS NOT MERELY LOCAL BIAS

In our formal concept learning experiments, we found similar results whether the prompt specified generating samples *'that may be from a fair coin with no correlation, or from some non-random algorithm'* or only *'from a fair coin, with 50% probability of Heads and 50% probability of Tails'*. This raises the question of whether the LLM is simply forgetting the prompt, and has a local text bias that leads it to be distracted by a long repeating pattern, instead of following the original task specification, as in neural text degeneration (Holtzman et al., 2019).

To test this alternative hypothesis, we used a real-world task used to evaluate few- and zero-shot learning in LLMs: the Massive Multitask Language Understanding (MMLU) dataset (Hendrycks et al., 2020). Each MMLU task includes a single multiple choice question, along with 4 multiple choice answers: A, B, C, and D. MMLU tasks are organized into subjects, intended to cover a broad range of common academic and professional subjects. We tested with two questions randomly selected from each of 5 subjects in MMLU: College Physics, Elementary Mathematics, High School Biology, High School Psychology, Philosophy.

Our experimental setup was such that we followed the MMLU original prompt format, which ends with the text *'Answer: ____'*. Although MMLU provides options for few-shot learning, we used zero context shots. We appended a sequence of flips *'Heads, Tails, Heads, Tails, ...'* from the concept $(01)^n$ to the end of this prompt, similar to how our experiments use a prompt format: *Q: ...  A: Heads, Tails, ...*.

An example full prompt for this experiment is: *'The following are multiple choice questions (with answers) about high school psychology.\n \n Nearsightedness results from \n A. too much curvature of the cornea and lens \n B. too little curvature of the cornea and lens \n C. too much curvature of the iris and lens \n D. too little curvature of the iris and lens \n Answer: Heads, Tails, Heads, Tails, Heads, Tails, Heads, Tails,'*.

As shown in Fig. 12, GPT-3.5 (`text-davinci-003`) shows dramatically different learning patterns in this MMLU setup, compared with our Randomness Generation results. In most MMLU questions, *GPT is extremely robust against local text bias*, maintaining a high probability of staying on-task and generating MMLU tokens even after hundreds of tokens (note: each flip, comma, and space together constitute 2-3 tokens) following a simple repeating pattern are appended to the prompt.

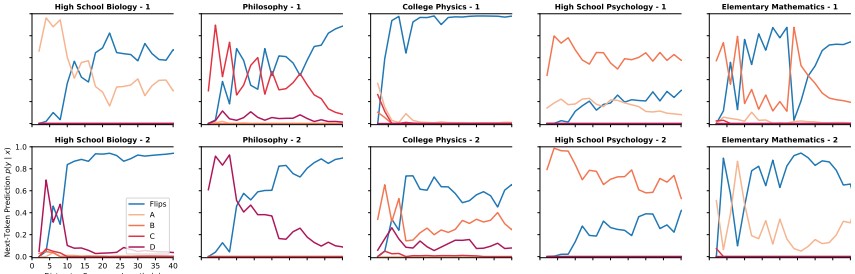

Figure 12: **GPT-3.5 demonstrates a task bias, returning to answer MMLU questions after many distractor flips** In these experiments, we take MMLU questions and append sequences of 'distractor flips', repeating samples from $(01)^n$, before querying the LLM for next-token probability. Individual lines are shown for flip tokens (*'Heads'*, *'Tails'*, *'T'*, *'ails'*) compared with MMLU answer tokens (*'A'*, *'B'*, *'C'*, *'D'*). Even with many distractor tokens, GPT-3.5 converges back to answering the MMLU question, demonstrating that our formal concept learning results for the Generation task are not merely a local text bias where the LLM forgets the original prompt.

The primary goal of this experiment was not to explore ICL dynamics in this MMLU distractor experiment, however, we do note striking patterns for different MMLU tasks. Though a few questions seem to demonstrate stable phase changes, sharply converging to high-confidence of generating flip tokens, we note that all but one (College Physics 1) of these converge to a state where MMLU tokens still have 5-20% probability. This probability means that, when temperature=1.0 and stochastic

decoding is used to sample completions, nearly all completions eventually converge back to the MMLU task, ending with an answer instead of more flips.

Note about Fig. 12: this shows the predictive probability $p(y|x = \ldots \text{', Tails')}$ for every other flip. With repeating $(01)^n$, the probability of sampling *'Tails'* after each *'Heads'* flip is very high, i.e. $p(y = \text{'Tails'} \mid x = \ldots \text{', Heads, ')} \approx 1$. While this is not always the case, we show $p(y|x)$ for every other flip to make important trends in learning dynamics more visible. In all of our analyses in the main text, we omit the probabilities of comma tokens $p(y = \text{','} \mid x)$ for similar reasons.

Although not shown here, we also found similar results appending distractor flips to prompts from GSM8k (`text-davinci-002` and `-003`), a simple mathematical benchmark commonly used to evaluate chain-of-thought reasoning. GPT's task bias appeared stronger in GSM8k, but was harder to evaluate, since GSM8k does not consist of multiple choice questions like MMLU.

## F    FORMAL LANGUAGE LEARNING WITH VARYING PREDICTION DEPTH

In our main text, for formal language learning with Generation tasks, we estimate the predictive probability $p(y \in C \mid x)$ by enumerating all possible binary sequences $y$ up to some depth $d$ and querying GPT for next-token prediction probabilities. In this appendix, we show results of this analysis with varying prediction depth $d$.

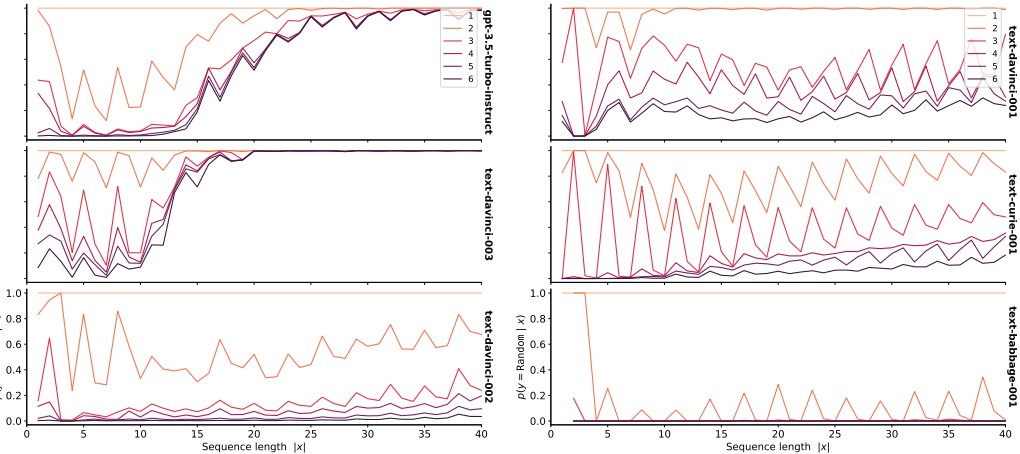

Figure 13: **Predictive distributions** $p(y|x)$ **by each LLM for Concept** $(\texttt{010})^n$**, at each prediction depth** $d$. Colors correspond to different prediction depths, also refer to Figure 5. Note: `text-ada-001` results are not shown since results did not follow the required format (*'Heads, Tails, . . . '*) adequately to be analyzed.

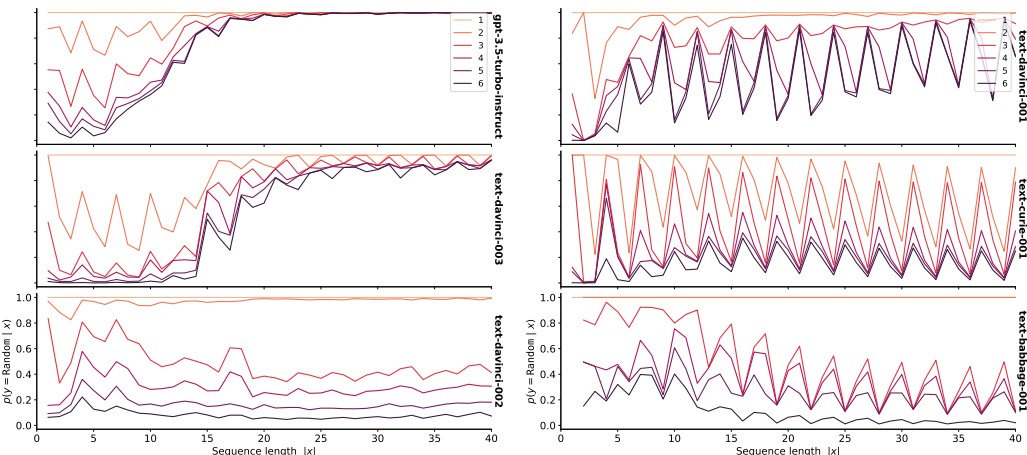

Figure 14: **Predictive distributions** $p(y|x)$ **by each LLM for Concept** $(\texttt{011})^n$**, at each prediction depth** $d$. Colors correspond to different prediction depths, also refer to Figure 5.

A notable takeaway of Figures 13, 14 is that with shallower depths ($d < 4$), we do not observe as stable ICL dynamics. Our predictive probability method is similar to a simple Markov Chain, where a greater depth $d$ similar to a higher Markov Chain order $k$, enables more expressive representation of $p(y|x)$. If $d = 2$, for example, then we only consider the 4 conditional probabilities $p(y = 0|x = 0)$, $p(y = 0|x = 1)$, $p(y = 1|x = 0)$, $p(y = 1|x = 1)$. With our methods of computing the predictive distribution, there are only three strings $y^*$ such that $y^* \in C$ for any depth, where at a depth $d$ there will be $2^d$ possible strings $y$. We also note that the predictive distributions $p(y|x)$ are increasingly close together, suggesting that with $d > 6$, the trends we observe will only change slightly.

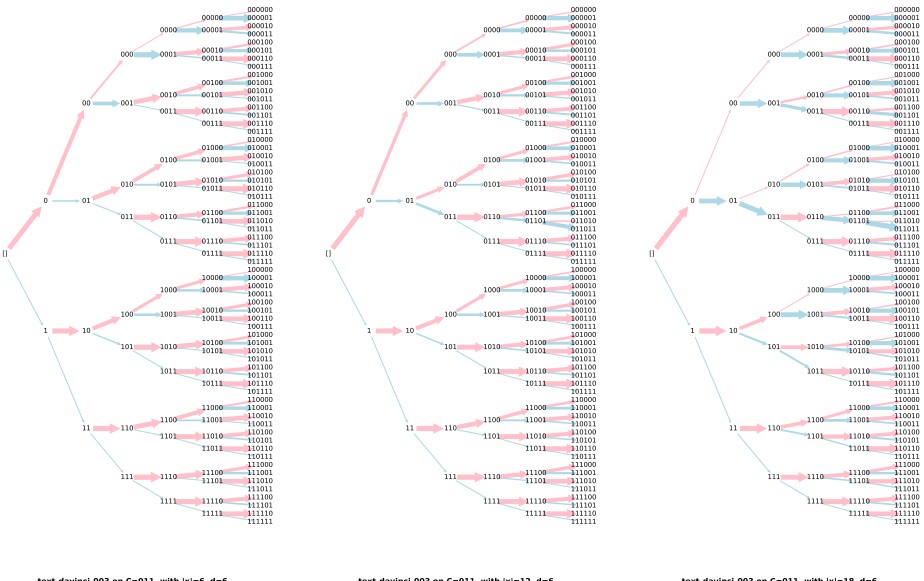

text-davinci-003 on C=011, with |x|=6  d=6     text-davinci-003 on C=011, with |x|=12  d=6     text-davinci-003 on C=011, with |x|=18  d=6

Figure 15: **Predictive distribution** $p(y|x)$ **trees with** $d = 6$ **for concept** $C = (011)^n$ **with** $|x| \in \{6, 12, 18\}$. Since $|x|$ is increasing by the same depth as the tree $\Delta_{|x|} = d = 6$, the transition from generating subjectively random numbers to deterministically repeating $011$ is visibly apparent. Also see Figure 5.

In Figures 15, 16, we show predictive distribution $p(y|x)$ trees, as in Figure 5 in the main text. Figure 15 should be read left-to-right, and Figure 16 should be read top-to-bottom. For example, in 15, the middle figure begins $x = 011011011011$ where the left figure ends ($x = 011011011011, y = 011011011011$), conditioned on the outcome sequence $y = 011011$. The right-most figure similarly begins where the left one ends.

Like Borges' Garden of Forking Paths (Bottou & Schölkopf, 2023), at each step in context (i.e. at some $|x|$ for a given $x$), LLM text generation presents the user with branching paths of possible token sequences. Some of these paths may lead to very different outcomes, in our case, either deterministically repeating some pattern or generating random flips. The domain of binary sequences lets us deeply explore and easily visualize a minimal version of this.

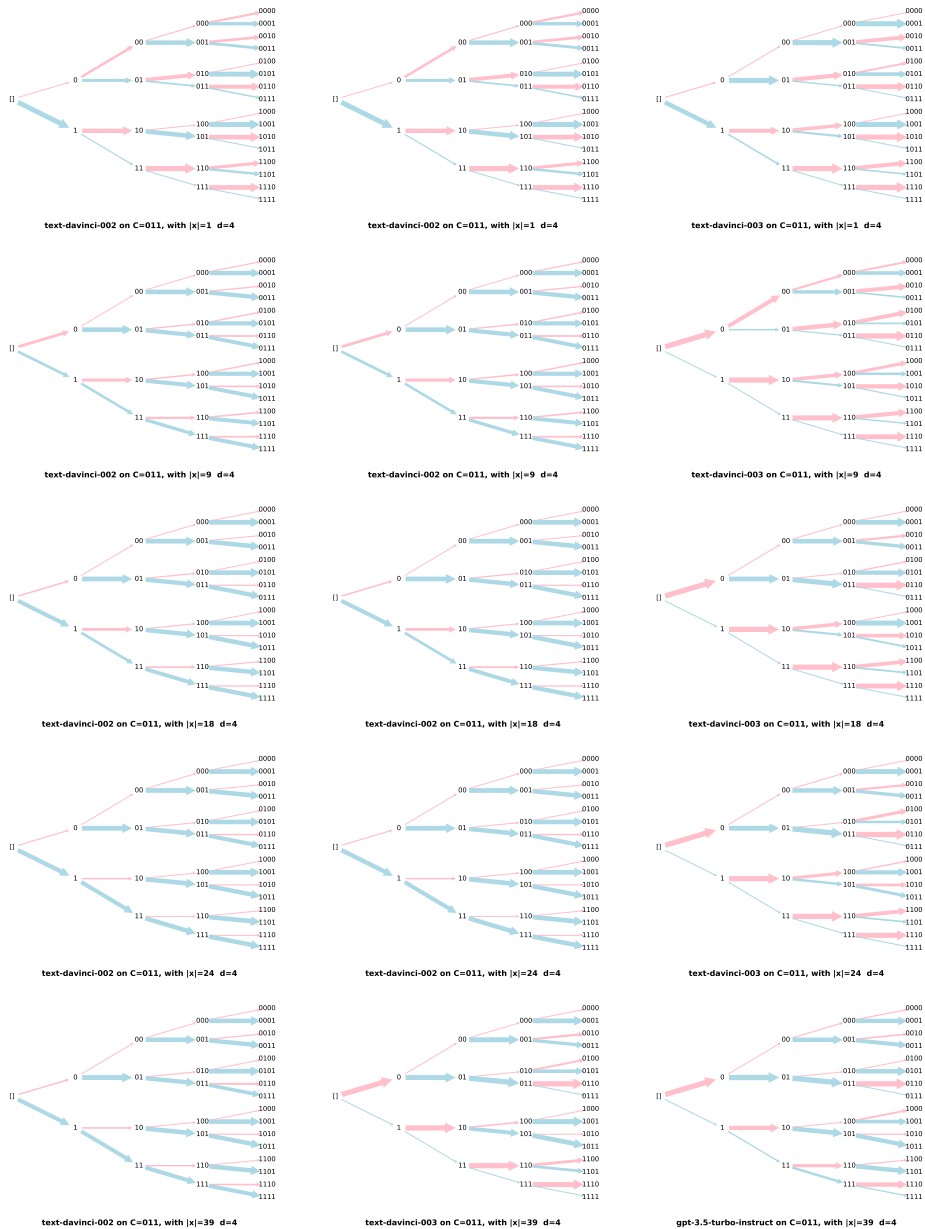

Figure 16: **Predictive distribution** $p(y|x)$ **trees with** $d = 4$ **for concept** $C = (011)^n$ **with** $|x| \in \{1, 9, 18, 24, 39\}$. Models shown are text-davinci-002, text-davinci-003, and gpt-3.5-turbo-instruct. Also see Figure 5.

# G  RANDOMNESS JUDGMENTS

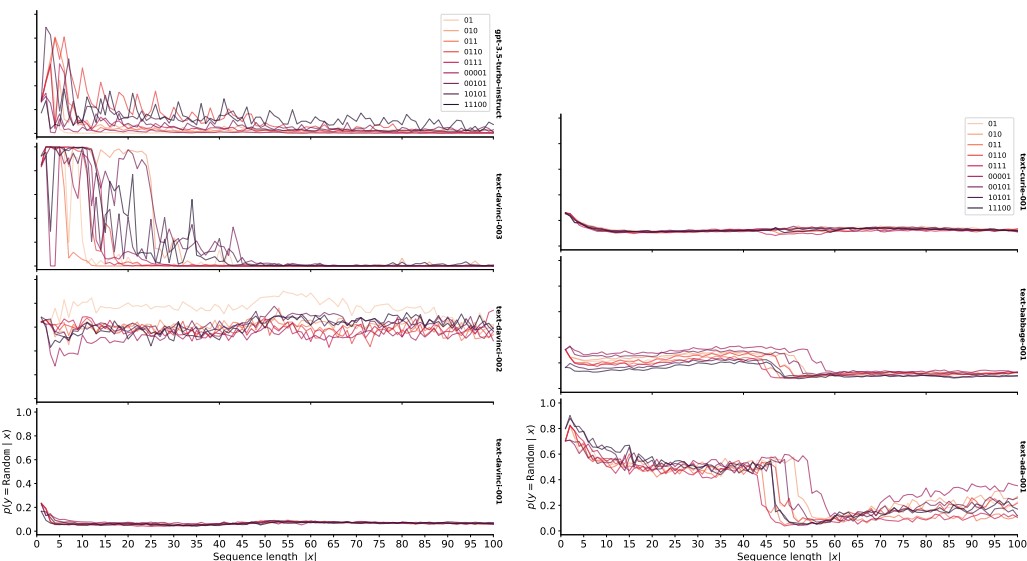

Figure 17: **Randomness judgments across OpenAI GPT models for 9 concepts**    Also see Fig. 4.    OpenAI models ordered from smallest/least fine tuned to largest/most: `text-ada-001` (??), `text-babbage-001` (1B params + SFT), `text-curie-001` (6.7B + SFT), `text-davinci-001` (175B + SFT), `text-davinci-002` (175B + SFT), `text-davinci-003` (175B + RLHF-PPO), `gpt-3.5-turbo-instruct` (??). These OpenAI model details were previously publicly available and have been archived at: https://archive. is/IXxGm. Note: technically `text-davinci-002` is a 'GPT-3.5' model, but in our text we use 'GPT-3.5+' to refer to `text-davinci-003` and later GPTs.

(Fig 17) `text-davinci-003` shows a stable pattern of being highly confident (high token probability) in the process being random up to some amount of context $|x|$, at which point it rapidly transitions to being highly confident in the process being non-random, with transition points varying substantially between concepts.  `chat-gpt-3.5-instruct` does not go through a stable high-confidence random period like `text-davinci-003`, and stable high-to-low confidence dynamics are observed for only a subset of concepts. The majority of earlier GPT models (`text-davinci-002`, `text-davinci-001`, `text-curie-001`, `text-babbage-001`) show no 'formal language learning', at all. However, surprisingly OpenAI's smallest available GPT model `text-ada-001` shows S-shaped in-context learning dynamics, with the peak close to .5 instead of 1.0 as in `text-davinci-003`. Additionally, the learning dynamics and transition points for all concepts appear nearly identical, approximately at $|x| = 50$, and some concepts show less stable "non-random" patterns for larger $|x|$.

LONGER FORMAL CONCEPTS

Testing `text-davinci-003` with longer concepts (Fig.19), many concepts still follow a pattern of sharply transitioning from high confidence in $p(y = \text{random}|x)$ to high confidence in $p(y = \text{non-random}|x)$. With increasingly long concepts, this pattern further degrades. This degradation is not surprising, for two reasons: (a) large concepts are more costly to represent, and so may either be more difficult for the LLM to learn, or the learning curves that show these patterns may be much wider (e.g. $|x|$ ranging from 0 to 1000 instead of 0 to 100, as we use). (b) Large concepts are less frequent in the training data, and so may be more prone to memorizing samples. When only a few samples are available, neural networks should be expected to memorize data points and context (Feldman, 2020), since there is not enough context to represent that pattern as a generic formal concept.

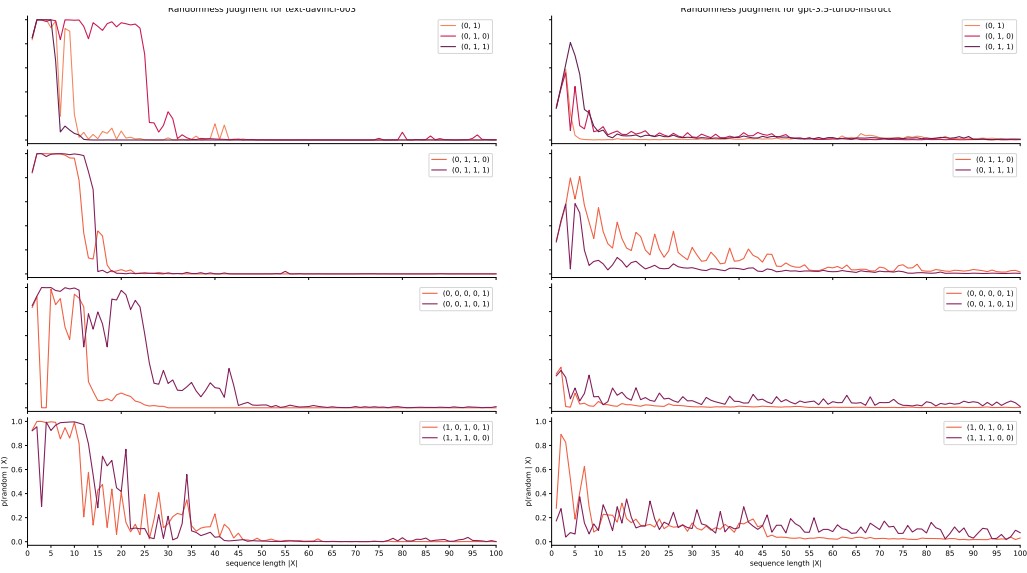

Figure 18: Randomness Judgment $p(y = \texttt{random}|x)$ dynamics for additional concepts tested, for `text-davinci-003` (Left) and `gpt-3.5-turbo-instruct` (Right).

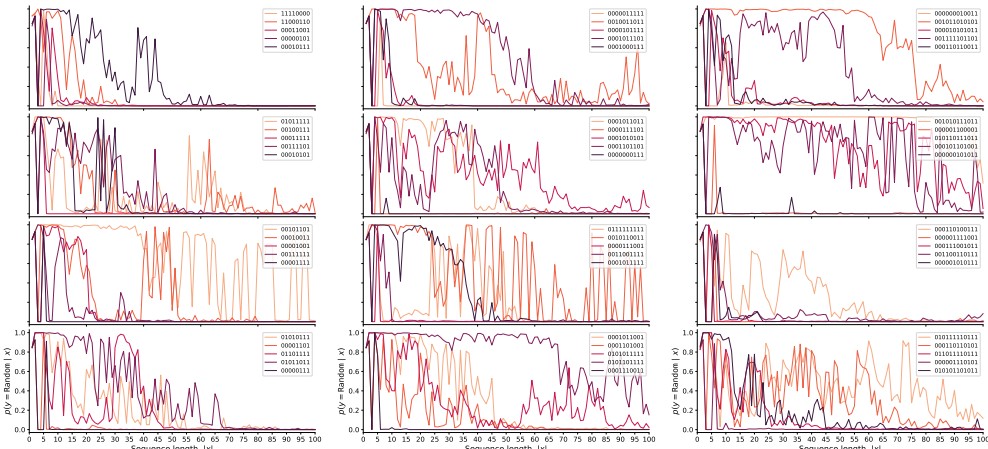

Figure 19: **Randomness Judgment with longer formal concepts** Similar to Fig. 4, we test Randomness Judgment with GPT-3.5 using longer repeating concepts. From left to right, concept lengths are: 8, 10, 12. Legends list colors corresponding to individual concepts.

RANDOM SEQUENCE BASELINE

To compare whether the patterns we find for Randomness Judgment are simply a function of increasing context length $|x|$ for *any* possible sequence, we test `text-davinci-003` with the Judgment task using $x \sim Bernoulli(.5)$ (Fig. 20). As a fair comparison to our formal concept learning task, we use the same 20 random sequences $x$ with varying length $|x|$, where each sequence corresponds to a single line and color.

While there is a higher overall average $p(Random|x)$ (blue line) for small $|x|$, with larger $|x|$ the average $p(Random|x)$ remains flat around 0.6. Contrast this with the formal concepts in Fig.4, where with large $|x|$, $p(Random|x)$ drops to nearly 0.0 .

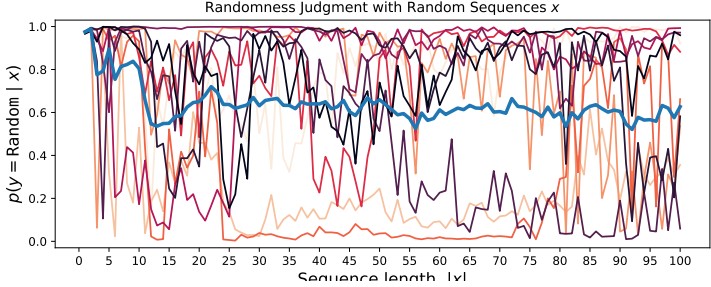

Figure 20: **Random baseline for Randomness Judgment task** As a baseline to compare with our concept learning $x$ sequences, we used a set of 20 sampled sequences $x \sim Bernoulli(.5)$. We varied $|x|$ along these individual sequences $x$, which enables us to consider ICL dynamics for individual sequences, represented here as lines.

## H  Random Sequence Generation by GPT Model

In Figures 21, 21, we show flip sequences $y$ from the Randomness Generation task, with various $p(\text{Tails})$ super-imposed on the same plot, with different colors according to the specified $p(\text{Tails})$. In Fig. 23, we compare the mean sequence probability $1/N \sum_i \overline{y^{(i)}}$, averaged over all GPT output sequences $y^{(i)}$, to the prompt-specified probabilities $p(\text{Tails})$.

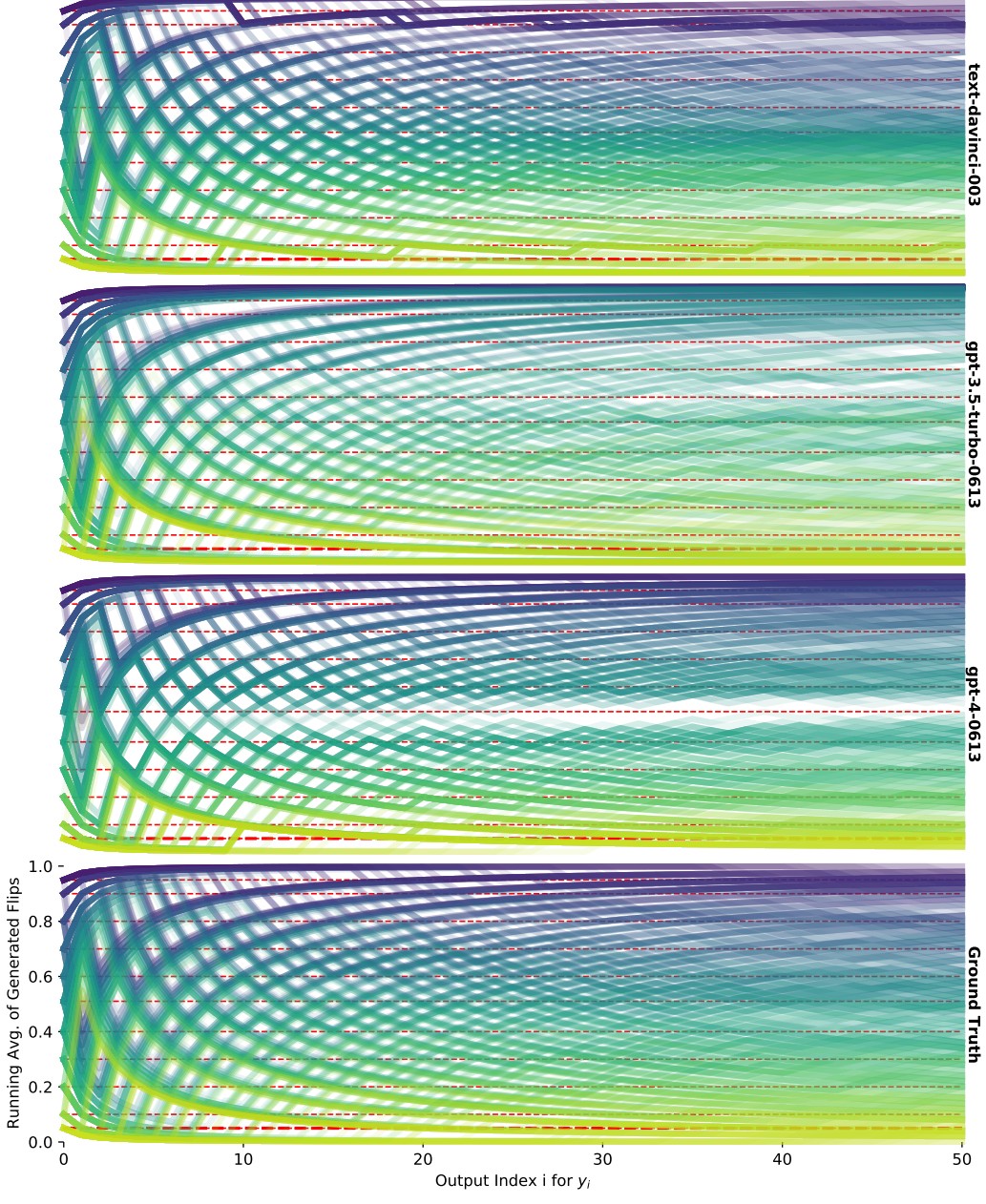

Figure 21: 50 sequences sampled by each GPT model, for each $p(Tails)$. Color is assigned according to specified $p(Tails)$. Red dotted lines are drawn for each $p(Tails)$.

Fig. 21 and the left side of Fig. 23 demonstrate that `text-davinci-003` and GPT-4 models not only are more controllable, following the correct probability more closely on average, but also have substantially lower variance than ChatGPT, which is both less controllable and has more variability

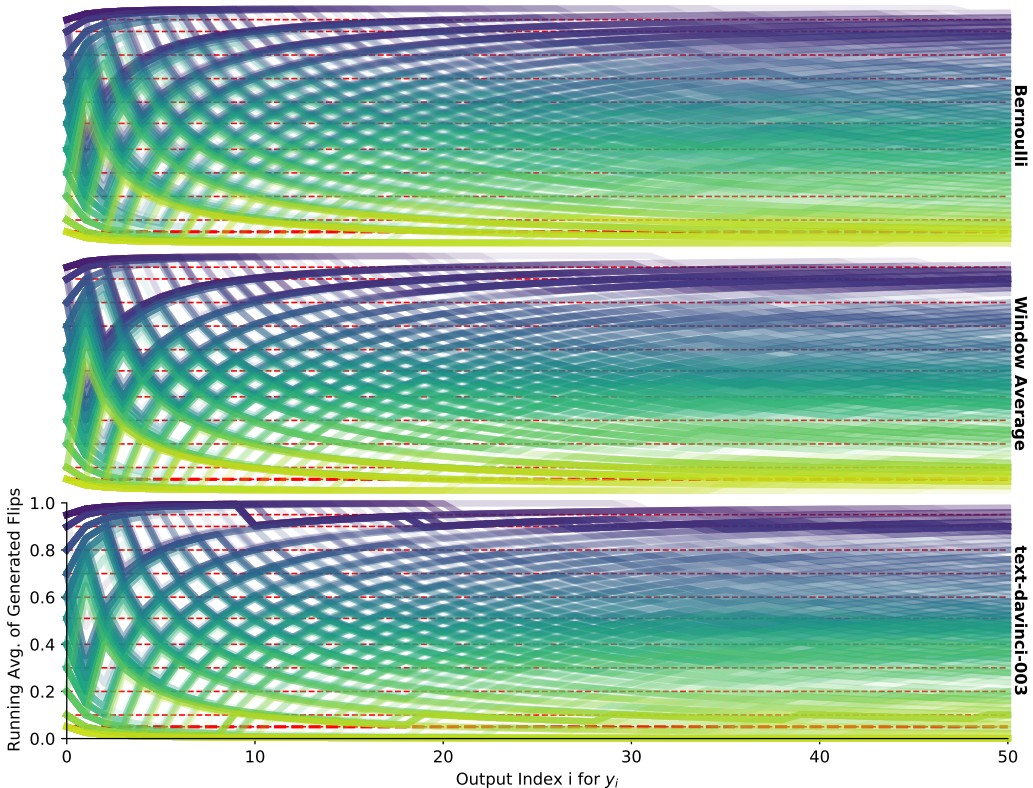

Figure 22: 50 sequences sampled by `text-davinci-003`, for each $p(Tails)$, compared with samples from Bernoulli and Window Average models fit to $y_{LLM}$ for each $p(Tails)$. Color is assigned according to specified $p(Tails)$.

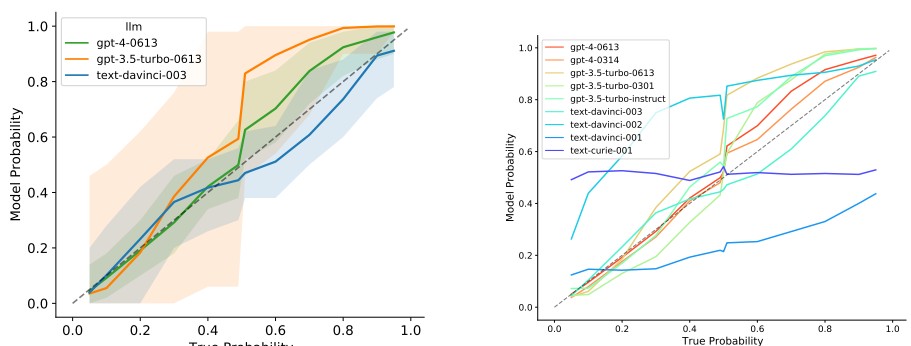

Figure 23: **Probability $p(\texttt{Tails})$ bias across LLMs** `text-davinci-003` and GPT-4 models are least biased relative to the specified $p(\texttt{Tails})$ (x-axis). In the left figure, error bars represent the maximum and minimum sequence means $\overline{y}$ for each $p(\texttt{Tails})$.

.

in its distribution of responses. Further, GPT-4 is lower variance than `text-davinci-003`, with sequences staying even closer to their means $\overline{y}$.

# I GAMBLER'S FALLACY METRICS BY GPT MODEL

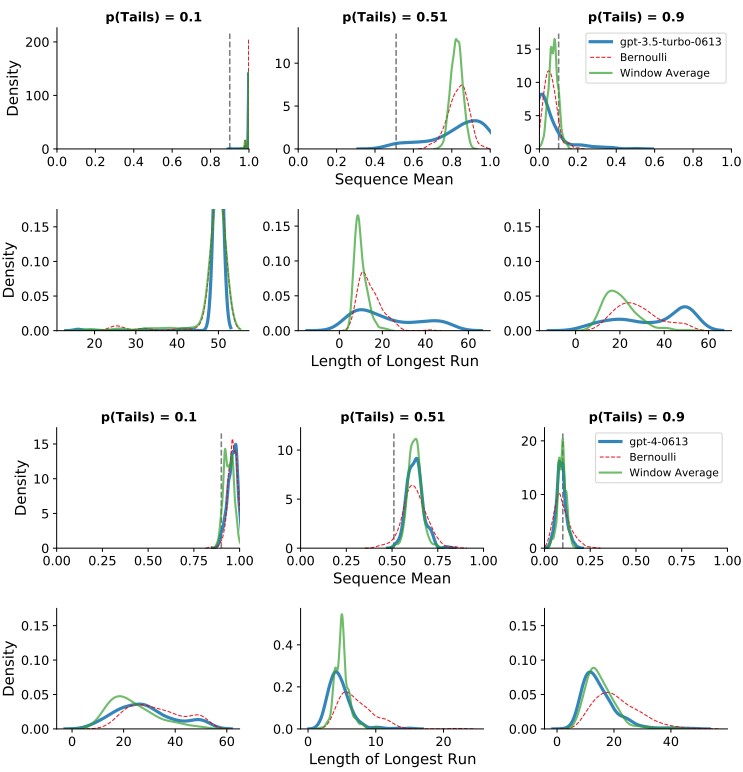

Figure 24: **Gambler's Fallacy histograms for ChatGPT (Top) and GPT-4 (Bottom).** Also see Fig. 7.

ChatGPT shows no clear Gambler's Fallacy bias, whereas GPT-4 does show this pattern, but is less pronounced than `text-davinci-003` (Fig. 24).

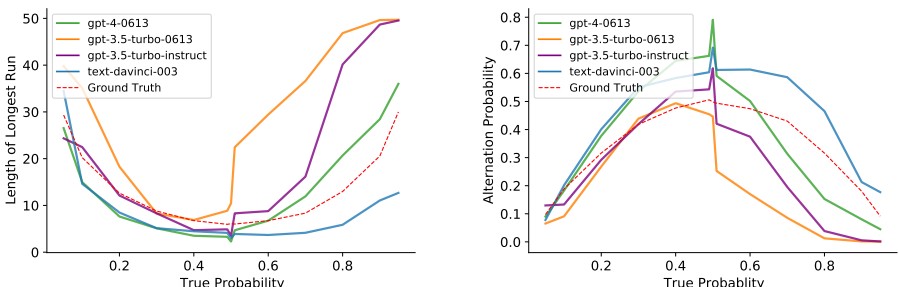

Figure 25: **Comparing metrics of Gambler's Fallacy across probabilities and LLMs** (Left) The mean longest run for each sequence $y$, at each specified probability $p(\texttt{Tails})$, where a run is a consecutive sub-sequence of the same flip repeating multiple times in a row. (Right) The mean alternation rate for each LLM, where alternation rate is the fraction of consecutive flips that are not equal $p(y_t \neq y_{t-1})$.

.

In both plots of Fig. 25, we observe that `text-davinci-003` shows a Gambler's Fallacy bias across $p(\texttt{Tails})$, of higher-than-chance alternation rates and shorter runs; ChatGPT (`gpt-3.5-turbo-0613`) produces more tails-biased and higher-variance sequences $y$ when $p(\texttt{Tails}) > 50\%$; GPT-4 and `gpt-3.5-turbo-instruct` interpolate between the two distinct

trends of `text-davinci-003` and ChatGPT. The red dotted line represents a Bernoulli process with mean $p(\texttt{Tails})$.

It is unclear how the capabilities we identify are implemented at a circuit level, or why they only seem to emerge in the most powerful and heavily tuned GPT models. For the latter, one hypothesis is that internet corpora contain text with human-generated or human-curating subjectively random binary sequences, and fine-tuning methods such as instruction fine-tuning, supervised fine-tuning, and RLHF make LLMs more controllable, enabling them to apply previously inaccessible capabilities in appropriate circumstances. Another hypothesis is that these fine-tuning methods bias LLMs towards non-repetitiveness, or induce some other general bias that plays a role in the in-context learning dynamics we observe in our particular domain. We hope that future work in cognitive and mechanistic interpretability will shed further light on these questions.

# J   MEMORIZATION, COMPRESSION, AND COMPLEXITY

Figure 26: **Distribution of unique sub-sequences for `text-davinci-003` for varying sub-sequence lengths** Also see Fig. 8.
.

In Fig. 27, we show that Markov chains of high order $k$ can account for the sub-sequence distribution, but this only applies when $k <= w$ where $w$ is the sub-sequence length, and the Markov chains can effectively memorizing the sub-sequence distribution of $y$.

Across both unnormalized and normalized distributions of unique sub-sequences (Fig. 28), we find that GPT-4 repeats the same length-10 sub-sequences significantly more than the other models, and both ChatGPT-based models (`gpt-3.5-turbo-0613`, `gpt-3.5-turbo-instruct`) follow different patterns for $p(\texttt{Tails}) < 50\%$ and $p(\texttt{Tails}) > 50\%$, even when controlling for sequence bias (Right). The only model that generates more unique sub-sequences than chance (above dotted line) is ChatGPT (`gpt-3.5-turbo-0613`).

As a coarse approximation of sequence complexity, we use Gzip file size of appended sequences $gzip(y : y' : y'' : \dots)$ and mean Levenshtein distance between sequences $d(y, y')$. Gzip (Deutsch, 1996), a common algorithm for file compression that is highly optimized for compressing strings with redundancy into small file sizes, and Gzip file size has been found to be an effective feature extractor for NLP (Jiang et al., 2022). Levenshtein distance Levenshtein et al. (1966) is a measure of edit distance between two strings.

Since sequence compression is highly correlated with probability, e.g. all sequences with $\overline{y} = 0.99$ will be highly compressible, we normalize the distribution of both plots in Fig. 29 by dividing by the same metric (appended Gzip size, or mean Levenshtein distance) for a Bernoulli distribution centered at $\overline{y}$. For all GPT models except ChatGPT, generated sequences have smaller Levenshtein distance than a Bernoulli process. This is evidence that these LLMs are using memorized sub-sequences

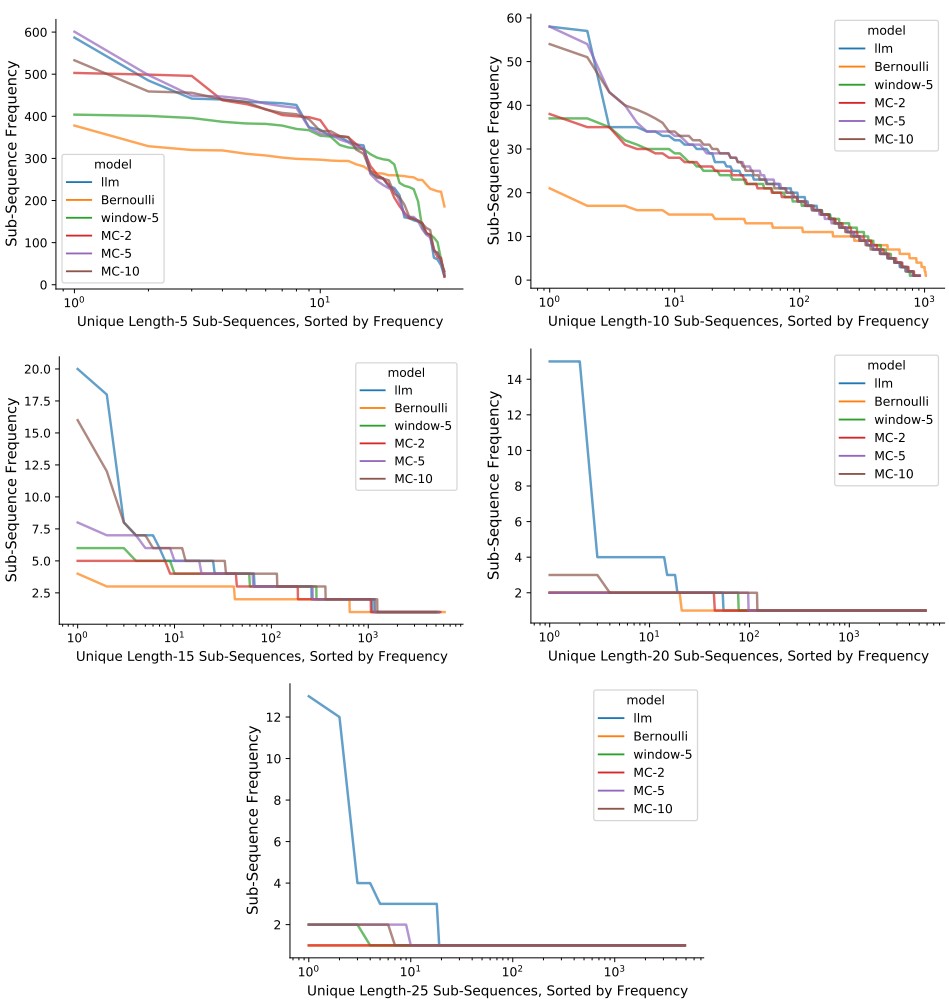

Figure 27: **Distribution of unique sub-sequences for `text-davinci-003`, with additional models, varying sub-sequence lengths** MC-2, MC-5, and MC-10 are Markov Chain models fit to GPT-3.5 flips, with orders $k = \{2, 5, 10\}$

.

('parroting'), since sequences have repeated structure. On the other hand, ChatGPT produces more dissimilar sequences than chance, suggesting *higher* complexity. In Gzip file size, however, we see a lower-complexity bias in all LLMs (except for a few higher values of $p(\texttt{Tails})$), to varying degrees, produce data $Y$ that is more compressible than data from an equal probability Bernoulli process.

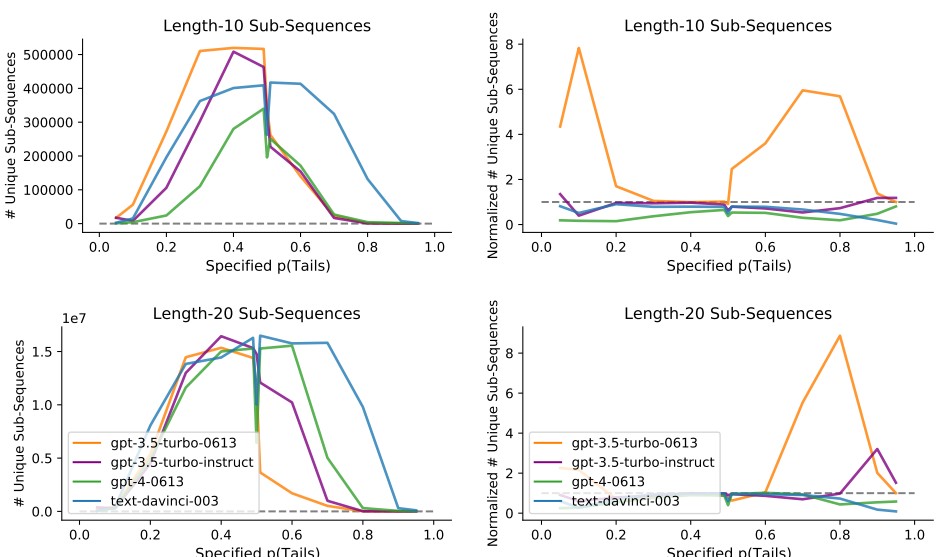

Figure 28: **GPT-4 repeats the same sub-sequences more often than other GPT models** (Left) Number of unique length-10 and length-20 sub-sequences as a function of specified probability $p(\texttt{Tails})$, across all sequences $y$ (note: $|y| = 50$) generated by each GPT model. (Right) The same distributions, with the y-axis normalized by dividing by the same metric (appended Gzip size, or mean Levenshtein distance) for a Bernoulli distribution centered at $\overline{y}$, to control for sequence compression being correlated with probability, e.g. with $\overline{y} = 0.99$, the same sub-sequences of only *'Tails'* flips will appear many times.

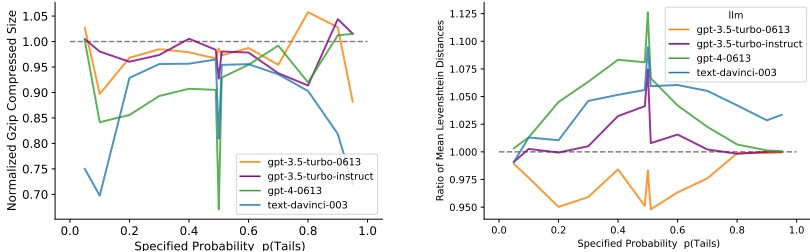

Figure 29: **GPT-generated sequences have lower complexity than Bernoulli sequences**

## K    BACKGROUND ON ALGORITHMIC AND SUBJECTIVE RANDOMNESS

Randomness of a sequence $x$, defined in terms of Bayesian model comparison between the class of non-random models with the class of random models, can be translated to be the difference between the sequence length $|x|$ and the algorithmic complexity, or *Kolmogorov complexity* of the sequence $K(x)$.

$$
\begin{aligned}
\text{randomness}(x) &= \log P(x|\text{random}) - \log P(x|\text{non-random}) \\
&= \log 2^{-|x|} - \log 2^{-K(x)} \\
&= K(x) - |x|
\end{aligned}
$$

The likelihood given a truly random Bernoulli process $p(x|\text{random}) = 2^{-|x|}$ since sequences of equal length have equal probability and there are $2^{|x|}$ binary sequences of length $|x|$. This can be thought of as a uniform prior over programs, where every program is an exact copy of the output string.

The likelihood of $x$ given the space of non-random processes marginalizes over the posterior of all non-random programs (hypotheses) $\mathcal{H}$:

$$
p(x|\text{non-random}) = \sum_{h \in \mathcal{H}} p(h)\, p(x|h)
$$

.

A natural prior for programs $p(h)$ is the description length of that program, where common metrics used in software engineering such as *lines of code* or *number of functions* can be seen as practical estimations of program description length.

If we assume $p(x|h)$ is a binary likelihood, that is:

$$
p(x|h) = \begin{cases} 1 & \text{if } h \text{ generates } x \\ 0 & \text{otherwise} \end{cases}
$$

and we simplify the problem to finding the maximum a-priori hypothesis $h$, and set a prior over hypotheses (programs) proportional to their length $p(h) = 2^{-|x|}$, this equates to finding the program with lowest Kolmogorov complexity $K(x)$:

$$
P(x|\text{non-random}) \approx \max_h p(h)\, p(x|h) = 2^{-K(x)}
$$

where Kolmogorov complexity $K(x)$ is defined as the description length of the shortest program that generates $x$ as output:

$$
K(x) = \operatorname*{argmin}_{\{p \in \Sigma^* | Evaluate(p) = x\}} |p|
$$

The notation $p \in \Sigma^*$ is analogous to $h \in \mathcal{H}$, but refers to a formal alphabet $\Sigma$ that programs are comprised of. In the general case, Kolmogorov complexity $K(x)$ is uncomputable due to the halting problem, since the expression $Evaluate(p) = x$ might run forever if $p$ has an infinite loop.

## L    OPEN SOURCE MODELS

We replicated our results across an array of open-source LLMs available through the HuggingFace Transformers API. Open-source LLMs are critical to replicability since many LLMs now available through commercial APIs may not be available in the future. **As of January 2024, the primary LLM analyzed in this paper `text-davinci-003` is no longer publicly accessible** [†].

In summary, we found similar results to GPT-3.5+ on Randomness Generation tasks varying $p(Tails)$ with LLMs with 50+ billion parameters (Fig. 32 & 33) and similar results on Randomness Judgment with `tulu-2-dpo-70b` (Fig. 30). These results supports our theory that with sufficiently large models and with reward-based fine-tuning (RLHF & DPO), LLMs can develop latent capabilities for (1) Bayesian model selection and (2) controllable generation of random binary sequence.

LLM inference was run on 2x NVIDIA A100 80gb GPUs in Harvard's FAS-RC cluster. Inference took a few days total GPU (2x) time with the Randomness Generation task. Randomness Judgment prompts were more efficient due to reduced number of output tokens.

We evaluated the following LLMs:

- `meta-llama/Llama-2-7b-hf`  (float32)
- `meta-llama/Llama-2-13b-hf`  (float32)
- `meta-llama/Llama-2-70b-hf`  (float16)
- `mistralai/Mistral-7B-v0.1`  (float32)
- `mistralai/Mistral-7B-Instruct-v0.1`  (float32)
- `mistralai/Mixtral-8x7B-v0.1`  (float16)
- `mistralai/Mixtral-8x7B-Instruct-v0.1`  (float16)
- `allenai/tulu-2-7b`  (float32)
- `allenai/tulu-2-13b`  (float32)
- `allenai/tulu-2-70b` [*]  (float16)
- `allenai/tulu-2-dpo-7b`  (float32)
- `allenai/tulu-2-dpo-13b`  (float32)
- `allenai/tulu-2-dpo-70b` [*]  (float16)

Following the suggested input format [†], when prompting Tulu 2 models we replace the *Q: . . .*    \n \n *A: . . .* prompt format used with other models with *<|user|>* \n . . .    \n \n    *<|assistant|>* \n . . . .

---

[†] https://openai.com/blog/gpt-4-api-general-availability
[†] https://huggingface.co/allenai/tulu-2-7b#input-format

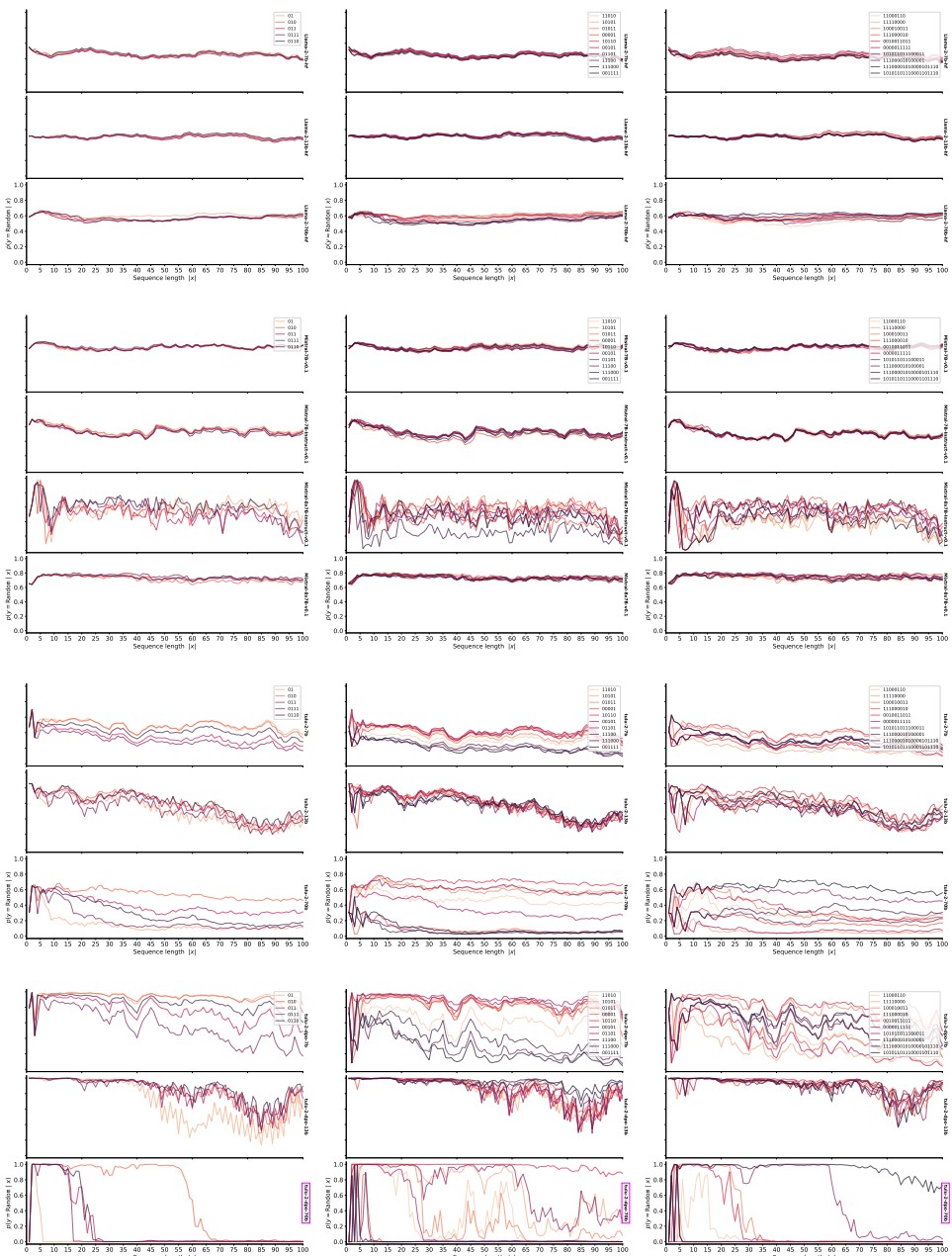

Figure 30: **Randomness judgments by open source LLMs – sharp transitions for Tulu 2 70b +DPO**    Similar to our results for smaller OpenAI GPT models (Fig. 17), we find relatively flat in-context learning curves for smaller open-source models and larger ones without RLHF or DPO. With `tulu-2-dpo-70b` we find sharp S-shaped learning curves, from high confidence in *Random* to high confidence in *Non-Random*, similar to our main results (Fig. 4 &  18) with larger and reward-optimized OpenAI models such as `text-davinci-003` and `gpt-3.5-turbo-instruct`. This supports our theory that with sufficiently large models and sufficient reward-based fine-tuning LLMs may develop latent Bayesian model selection capabilities.

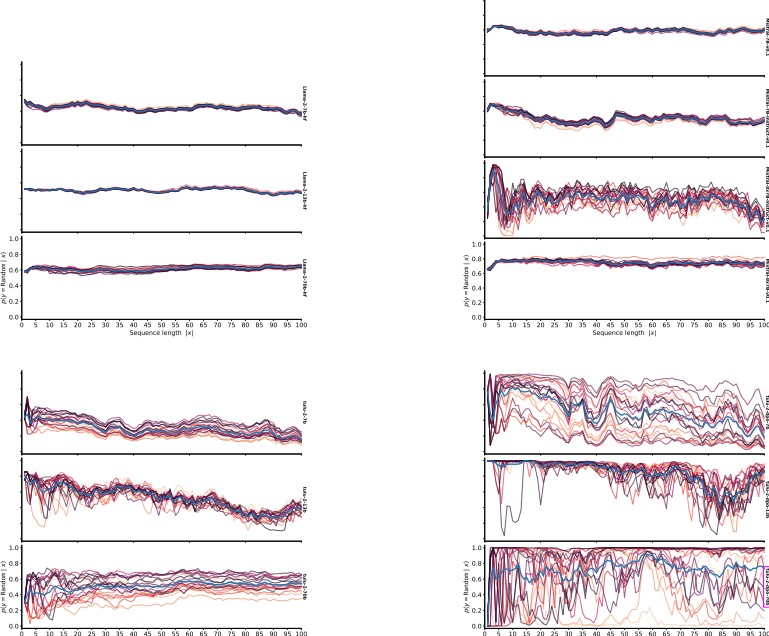

Figure 31: **Randomness judgments by open source models for random input sequences** $x$
Following Fig. 30 and 20, we provide as a baseline for open-source models, Randomness Judgment
on random sequences $x$. Here we show 20 random concepts $x$. Notably, for `tulu-2-dpo-70b` we
do not see the same sharp S-shaped ICL curves as in Fig. 30.

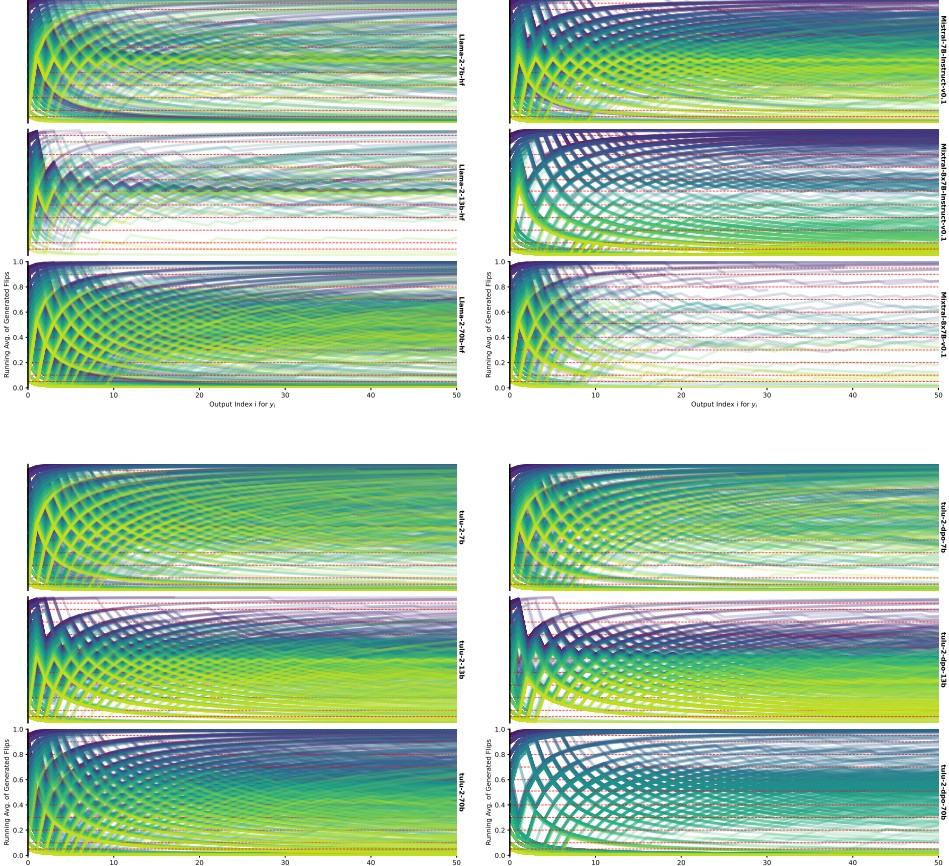

Figure 32: **Random flip generation flips with** $p(Tails)$ **specified by each open-source LLM**
Random flip generation running averages for all specified $p(Tails)$, where color and red dotted lines
correspond to different $p(Tails)$. Similar to Fig. 21 we find that larger LLMs, particularly Tulu 2 70b
models, demonstrate the capability of generating random binary sequences with a controllable mean
$p(Tails)$. These plots show 100 samples for each model and $p(Tails)$. `Mistral-7B-v0.1` is
excluded since we found its flip generation was not reliably formatted.

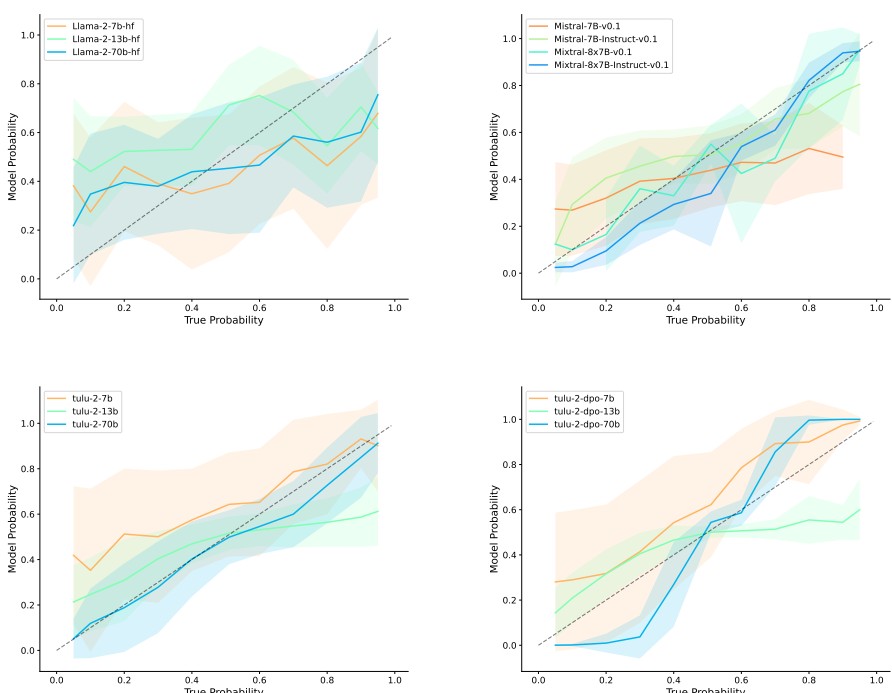

Figure 33: **Probability bias for each open source LLM** Bias between LLM-generated flip sequence means $\overline{x}$ compared to specified $p(Tails)$. All tulu models are controllable as well as all Mistral models except `Mistral-7B-v0.1`. Error bars shown for 1 standard deviation.

