# OpenReview forum: "In-Context Learning Dynamics with Random Binary Sequences"
_ICLR.cc/2024/Conference — ICLR 2024 poster_

### Official Review · Reviewer_rqBo · 2023-10-18

**Soundness:** 2 fair
**Presentation:** 3 good
**Contribution:** 3 good
**Rating:** 6
**Confidence:** 4

**Summary:**

This paper proposes a cognitive interpretability framework (IMO, a Bayesian model selection framework similar to those used in iterated learning or rational speech model) to analyze the dynamics of LLM conducting in-context learning (ICL). The analysis can help us understand what the latent concept is (it seems hard to understand this term without reading [1]) in ICL. Specifically, the paper considers a random binary sequence-generating problem and GPT series models for experiments. They find the model exhibits very interesting behaviors, e.g., it matches the Bayesian hypothesis selection process well; there is a sharp phase change when manipulating the prompting input length, indicating specific hypothesis becomes dominant, etc. Although the paper studies a very important problem by combining results in different fields, I cannot give a clear-cut conclusion of what is the core contribution of this paper. It would be helpful if the authors could highlight some core concepts in a relatively simple way and explain the results more. Also, I have several concerns about the technique details of the analysis (see the weakness part). After reading the paper several times, I believe the paper has big potential. I would give the current version a 3, but I would be happy to increase my score if my concerns are tackled during the rebuttal.

[1] Xie, Sang Michael, et al. "An explanation of in-context learning as implicit Bayesian inference." ICLR 2022

**Strengths:**

This is an interesting interdisciplinary study of deep learning and cognitive science, which is quite novel to me. The results of the simple binary sequence are quite persuasive.

**Weaknesses:**

1. The paper is very hard to follow. Section 1, 2, and 3 provides the background, preliminaries, and the system setting of the paper, but I find it hard to extract the system setting of the paper. Most of the text in this part discusses some insights and hypotheses from related works in these two fields. It would be helpful to squeeze this part and discuss the related works in another section.
2. The results on binary sequence are good, but it is not sure whether the findings can be extended to other realistic tasks. It would be helpful if the author could verify these findings in more real datasets, e.g., math problems, question answering, etc.
3. There is not so much discussion about the Bayesian model. Going deeper in this direction will help the paper a lot.

**Questions:**

1. For Figure 2, what do x (green), LLM (yellow), and ICL (purple) mean? There is no such color in the figure.
2. I feel confused about the notation $p(y=Random | x)$, I guess $y\in[ H, T ]^n$ or $y\in [ 0, 1 ]^n$ is the output sequence. Should it be $p(h=Random | x)$?
3. In section 4, the notation of $\bar{x}_ {t-w…t}$ in $p(y|x)$ is not quite clear. I guess it should be $\bar{x}_ {t-w,…,t}$.
4. Figures like Figure-5-right are quite hard to understand. What is the take-away message? Is this saying GPT performs more similar to the window average model and less similar to the Bernoulli model? (in terms of how wide the curves spread around the converged mean)
5. In the last paragraph of section 4, what is the $C$ in $x=C^{|x|}$ and $y_t\in C$ means. Where is $C$ defined? (I can find $c$ and $\mathcal{C}$ in section 2, but cannot find $C$).

---

> ### Author Response · Authors · 2023-11-19
>
> **(Part 1/3)**
>
>
>
> Thank you for the helpful suggestions and feedback! We are very glad that you see this work as “tackling an important problem”, our results to be “very interesting” and “quite persuasive”, and that you believe it has “big potential”. We respond to your specific comments below.
>
>
> > This paper proposes a cognitive interpretability framework (IMO, a Bayesian model selection framework similar to those used in iterated learning or rational speech model) to analyze the dynamics of LLM conducting in-context learning (ICL). The analysis can help us understand what the latent concept is (it seems hard to understand this term without reading [1*]) in ICL.
> >
> > [1*] Xie, Sang Michael, et al. "An explanation of in-context learning as implicit Bayesian inference." ICLR 2022
>
> This approach indeed is similar to iterated learning and RSA in that we consider LLM behavior in a Bayesian model selection framework. However, our work differs from prior work such as [0] that apply iterated learning or RSA as techniques to improve LLM abilities. More similar to the evolutionary dynamics of [1], or the Bayesian concept learning dynamics of [2, 3] we study in-context learning dynamics. To our knowledge, our work is novel in being the first to systematically analyze dynamics of ICL over different input ($x$, in our notation) and output ($y$) token sequences. We have modified Section 2 to more clearly explain our framework without requiring reading [1*], and provide further discussion of Cognitive Interpretability in Appendix C.
>
> [0] Zelikman, E., Mu, J., Goodman, N. D., & Wu, Y. T. (2022). Star: Self-taught reasoner bootstrapping reasoning with reasoning.
>
> [1] Kirby, S., & Hurford, J. R. (2002). The emergence of linguistic structure: An overview of the iterated learning model.
>
> [2] Ullman, T. D., Goodman, N. D., & Tenenbaum, J. B. (2012). Theory learning as stochastic search in the language of thought.
>
> [3] Piantadosi, S. T., Tenenbaum, J. B., & Goodman, N. D. (2012). Bootstrapping in a language of thought: A formal model of numerical concept learning
>
>
> > I cannot give a clear-cut conclusion of what is the core contribution of this paper. It would be helpful if the authors could highlight some core concepts in a relatively simple way and explain the results more.
>
> Thank you for raising this question. We first summarize our core contributions as follows:
> - **Evaluating ICL as Bayesian model selection in a practical setting** (Sec. 2). We find evidence of in-context learning operating as Bayesian model selection in LLMs in the wild, compared to prior work that trains smaller toy neural networks from scratch. This grounds past theoretical work, which used small neural networks trained from scratch, in simple phenomena observed with pre-trained blackbox LLMs, and it serves as a first step towards systematically analyzing in-context learning dynamics in LLMs and comparing text generation dynamics to simpler (cognitively inspired, in our case) models.
> - **Subjective Randomness as a domain for studying ICL dynamics** (Sec. 3). We introduce a new domain, subjective randomness, which is rich enough to be theoretically interesting, but simple enough to enable analysis of complex behavioral patterns. Traditional few-shot learning domains are difficult to analyze because of the vast number of tokens that can be sampled at each step of ICL and token generation (i.e. Fig. 5 would be impossible to visualize), whereas subjective randomness simplifies text generation to a two-token format where learning dynamics are more apparent.
> - **Sharp transitions in abilities to in-context learning dynamics** (Fig. 4, 5).  We systematically analyze in-context learning dynamics in LLMs without observing or updating model weights, demonstrating sharp phase-changes in model behavior. This is a minimal, interpretable example of how the largest and most heavily fine-tuned LLMs can suddenly shift from one pattern of behavior to another during text generation, and supports theories of ICL as model selection.
>
>  We added a new illustrative toy example with Figure 2 (also see Appendix D), to help readers understand the third point, and our claim that sharp transitions in the predictive distribution (i.e. token-level behavior) support that ICL is serving as model selection. Along with this toy example, we updated the text in Sections 2 and 3 to better explain these concepts as well as our method for evaluating formal concept learning in Generation tasks. We have also updated text accompanying each key result, summarizing our interpretation and how this result supports our claims.
>
>
>
>
> *(continued below)*

---

> ### Author Response · Authors · 2023-11-19
>
> **(Part 2/3)**
>
>
> > The paper is very hard to follow. Section 1, 2, and 3 provides the background, preliminaries, and the system setting of the paper, but I find it hard to extract the system setting of the paper. Most of the text in this part discusses some insights and hypotheses from related works in these two fields. It would be helpful to squeeze this part and discuss the related works in another section.
>
> Thank you for the suggestions. We have trimmed down the first two introductory sections of our paper. Instead, we place more emphasis on connecting our theories with our results, and describing the significance of both.
>
>
> > The results on binary sequence are good, but it is not sure whether the findings can be extended to other realistic tasks. It would be helpful if the author could verify these findings in more real datasets, e.g., math problems, question answering, etc.
>
> We have added an additional sets of experiments which may help this address this question: (a) appending binary sequences to zero-shot learning prompts for MMLU and testing whether the LLM changes tasks to generating coin flips (see Appendix E), and (b) Randomness Judgment tasks with longer length formal concepts. In a response to Reviewer F7kD, we describe challenges with using more complex concepts. For (a), though thorough analysis of ICL dynamics was not an intended goal of the MMLU experiment, we do find interesting non-linear ICL dynamics in a domain that is closer to a real-world task.
>
> Additionally, we added new text, particularly Section 2, explaining the significance of our findings of ICL phase changes and model selection. Our work is the first such demonstration that we know of, and we see enormous opportunity in future research that builds on our work, analyzing in-context learning dynamics in domains typically used for few-shot learning in math problems and question-answering. The sorts of sharp transitions in ICL we observe may be common in other few-shot learning domains, meaning that with just one or two more in-context samples, model behavior might change suddenly from one pattern to another. We see this work as an initial consideration and demonstration of this widely impactful principle for ICL.
>
>
> > There is not so much discussion about the Bayesian model. Going deeper in this direction will help the paper a lot.
>
> Thank you for this suggestion! We have significantly improved discussion of the Bayesian model-–please see updated Sections 2 and 4. We have also added a new figure with an illustrative example, Figure 2, and an accompanying Appendix D, to help the reader connect our claims with our empirical results. We have also improved discussion of our Bayesian methodology for analyzing the LLM posterior predictive distribution with formal concept learning in flip Generation tasks (end of Section 4).
>
>
>
>
> *(continued below)*

---

> ### Author Response · Authors · 2023-11-19
>
> **(Part 3/3)**
>
>
>
> > For Figure 2, what do x (green), LLM (yellow), and ICL (purple) mean? There is no such color in the figure.
>
> We apologize for any confusion here. This was an error due to a last-minute aesthetic change with Figure 1 (previously Figure 2), and the caption has been fixed now. Following the figure labels on the right-hand side, $x$ is the input (top, blue), the LLM’s latent concept space is represented by the middle box. In the middle box, latent concepts selected by ICL are red and have a shaded box behind them.
>
>
> > I feel confused about the notation p(y=Random|x), I guess y∈[H,T]n or y∈[0,1]n is the output sequence. Should it be p(h=Random|x)?
>
> We note that the current notation, p(y = Random | x), is actually correct. Though we do indeed consider Random and Non-Random hypotheses, our main metric is the predictive distribution p(y | x) of output tokens $y$. For Randomness Judgment tasks, we measure the probabilities for the tokens $y$ "*Random*" and "*Non*" (-Random), as in the prompt *"... The sequence was generated by a \_\_\_\_\_"*. In Randomness Generation tasks, $y$ instead represents token sequences as you suggest, as in the prompt: "*Generate a sequence of 1000 random samples … A: [Heads, Tails, \_\_\_\_\_*".
>
>
> > In section 4, the notation of x¯t−w…t   in  p(y|x)  is not quite clear. I guess it should be x¯t−w,…,t
>
> Thank you for this feedback! We have updated the notation in this formula to include commas, as well as other formulas with similar notation.
>
>
>   > Figures like Figure-5-right are quite hard to understand. What is the take-away message? Is this saying GPT performs more similar to the window average model and less similar to the Bernoulli model?
>
> Yes! That indeed is one key takeaway message. A second message is that GPT is not simply outputting the same memorized sequences–-instead, we are seeing evidence of an interesting, algorithmic capability. Each text generation sequence has a unique trajectory, as demonstrated by the running average lines, and each path stays relatively close to the sequence mean. If flip sequences relied on simple memorization, then certain running average trajectories would be much darker than others. To improve legibility and address your concerns, we note that we have also edited the text around sections 4 and 5 to better explain this point, though due to space limitations, we cannot mention it in this figure’s caption (now Figure 6).
>
> A third message is that, when looking at the figure, we immediately see interesting *dynamics* over text generation. Most ICL and few-shot learning work assumes that additional context, or shots, lead to stable and consistent improvements in accuracy. In contrast to that view, here we see sequences of tokens with interesting token-level behavioral dynamics.
>
>
>
>
> > In the last paragraph of section 4, what is the C in x=C|x| and yt∈C means. Where is C defined? (I can find c and C in section 2, but cannot find C).
>
> Following commonly used notation used in Bayesian modeling to define posterior predictive distributions, $C$ is the concept pattern, for example, if $C = (011)^n$ and $y = 01101101$, then $y \in C$. To address your concerns, we have edited this part of the text to clarify this definition, as well as the related notation $p(y \in C | x)$.
>
>
>
> **Rebuttals Summary**: Thank you again for your feedback! Following your suggestions, we have made changes to our submission, including trimming the introductory sections of our writing and devoting more text (e.g. at the end of section 2 and the end of section 4) as well as an introductory figure (Figure 2) to helping readers understand our Bayesian modeling framework. To your point about wondering what the broader impact of this work is, we further emphasize the point that the phase changes we observe in ICL may be present or even common in real-world few-shot learning domains as additional “shots” are provided for a given prompt (longer $|x|$, in our notation), particularly when individual data sequences $x$ are considered rather than averaging over many samples. Additionally, we have added new experiments to our Appendix, including an experiment (Appendix E) which injects “Heads, Tails, …” flip sequences into MMLU task. Though it was not the goal of this experiment, we observe compelling in-context learning curves, initial evidence that we may find complex, non-linear ICL dynamics in “realistic” few-shot learning tasks. We hope our changes and individual responses to your points address your raised concerns and hope that you will consider raising your score to support the acceptance of our work.

---

> ### Comment · Reviewer_rqBo · 2023-11-19
> **Thanks for the rebuttal, the paper improves a lot and my major concerns are resolved, hence I raise my score to 6.**
>
> Looking forward to seeing the code and camera-ready version.

---

### Official Review · Reviewer_F7kD · 2023-10-26

**Soundness:** 2 fair
**Presentation:** 3 good
**Contribution:** 3 good
**Rating:** 6
**Confidence:** 2

**Summary:**

This paper studies in-context learning (ICL) of LLMs with experiments inspired by subjective randomness.

Subjective randomness is concdididerned with the human perception of randomness. A main takeaway of this framework is that strings are more likely to be perceived as random if they can be generated by a short (deterministic) program. More formally, the subjective randomness of a string can be defined as the difference between the log-likelihood of it being generated by uniformly at random, and it being generated by a determinstic program. As the former quantity is fixed (for a given length), the subjective randomness is proportional to the Kolmogorov complexity of the string.

This paper presents experiments evaluating the ability of several GPT-3.5+ models (hereafter, GPT) to generate seemingly random strings, and distinguish uniformly random strings from deterministicly generated strings. These experiments are carried out by explicitly prompting the model to accomplish each task.

In the generation experiments, the authors find that the statistics of strings generated by GPT differ from uniformly random. For example, the distribution of the running average bit is tighter around the mean as compared to uniformly random strings (Figure 5); more specifically, GPT avoids long single-value runs, i.e., exhibits the gamber's fallacy (Figure 6). Furthermore, strings generated by GPT have simple sub-sequences (as measured by Gzip file size or Levenshtein (aka edit) distance).

For the distinguishing experiments, GPT is prompted with strings of increasing length, generated either uniformly at random or from a simple regular language (e.g. (01)^n, or (010)^n). The authors find that each language has a sharp phase shift in GPT's distinguishing ability: for example, when feeding contexts from (01)^n, the likelihood that GPT predicts the context to be determinstically-generated increases from ~0.5 (a random guess) at n=3, to ~0.8 at n=4.

**Strengths:**

- The paper tackles an important and formidable question: what is really happening in ICL? Considering the complexity and opacity of LLMs, this creative approach could make a welcome addition to the literature.
- Specifically, this paper is, to my knowledge, the first to propose subjective randomness as a method of analyzing LLMs. This is an original idea.
- This work can serve as an interdisciplinary bridge for computer scientists into cognitive science. As more works are concerned with the "cognitive abilities" of LLMs (and in general, adopt a humanizing/personifying lens on LLMs), it makes sense to draw from the existing rich literature developed in the field of cognitive science itself.
- As a reader with a computer science background (esp. pseudorandomness), I found the high-level introduction to subjective randomness to be well-written and accessible.
- The methods presented in this paper are black-box: they only require prompting access to the language model, but do not use the weights, architecture, or training data. This is increasingly important as not all researchers reveal these attributes upon release of an LLM.

**Weaknesses:**

In decreasing order of importance.
### Usage of the term "pseudorandomness"
The use of the term "pseudorandomness" in this paper differs significantly from how the term is commonly used in computer science literature. In particular, a distribution can only be pseudorandom with respect to a (computationally-bounded) class of distinguishers. However, throughout this paper, the ability of GPT to generate "pseudorandom" numbers is discussed without an explicit specification of a distinguisher. Perhaps the use of the term "pseudorandomness" is more appropriate in the experiments in which GPT is the distinguisher (rather than the generator), but there the generator itself is extremely trivial---it is a determinstic function, which therefiore can be distinguished by an almost-computatioanlly-trivial class of functions (that output 1 iff their input matches the (fixed!) output of the generator).

To avoid confusion with a long-standing and clearly-demarcated field of research, I suggest the authors to avoid using the term "pseudorandom" in this paper. Instead, they could use a term such as "subjectively-random" or even "seemingly-random", which avoids alluding to a concept that is not appropriate in this experimental setting.

### What do the experimental results actually mean?
A significant contribution of this paper is in demonstrating how subjective randomness can be used to experiment on LLMs. However, I am not sure what insight can be garnered from the particular experiments presented in this paper.
My understanding of the results from the experiments presented in the body of the paper is:
1. GPT does not generate uniformly random numbers; in particular, it exhibits the gambler's fallacy.
2. As context length increases, GPT can more confidently distinguish uniformy random strings from extremely simple, deterministically-generated strings (such as (01)^n).

I find neither of these conlcusions particularly insightful towards a deeper understanding of ICL dynamics. As a suggestion for improvement, I would expand the setup of the distinguishing experiments (item 2 above) along two axes:
- Stretch (length of generated string): Rather than testing on deterministically-generated strings (which are NOT pseudorandom in any non-trivial sense of the word), have GPT distinguish between uniformly random strings and strings generated by an actual pseudorandom generator; that is, one that takes as input a short random seed, and stretches it into a longer string.
- Generator complexity: The automaton for detecting the language used in the current experiments is only a handful of cells large. Yet, GPT has billions of parameters. If you insist on using deterministically-generated strings in the distinguishing experiments, at least have them be generated by significantly more complex programs.

One of the main contributions claimed in page 3 was that the subjective randomness framework will provide convincing evidence that ICL carries out a form of Bayesian model selection. I could not see how the experimental results I listed above support this claim. Please let me know if I am missing a connection here.

### Figures
- The colors in the caption of Figure 2 do not align with the figure itself. This is one of the main figures in the paper! On that note, I'd suggest using \color{} in the caption when referring to the different colors.

- Please review all figures in the body of the paper for colorblind-accessability. I am colorblind and had a hard time deciphering Figure 8. There are plenty of guides online on how to make accessible figures, and I am happy to point you to specific ones if that is needed.

- What does "Probability of Concept" mean on the y axis of Figure 8? My understanding is that it is the probability that, when prompted by a context of length $|x|$ from the regular language, the model responds that the input is deterministically-generated. Is that correct? (See also the last question in Questions section.)

### Writing on page 7
The mathematical notation in the sum on page 7 was confusing (the righthand side that following "and computing"): I didn't understand what is $(y_d == 1)^{(i)}$--though I may have some guesses, please give a full definition of notation you are using in your paper.

There is a typo in the same paragraph: "asses" should be "assess".

**Questions:**

- Thank you for introducing me to the area of subjective randomness. Can you confirm that my understanding of it is correct, as far as needed to understand this paper? Am I missing a key insight from this area?
- When talking about distinguishing ability, the correct quantity to look at is the "advantage" of the distinguisher: Denoting the uniform distribution, the pseudorandom generator, and the distinguisher by $U$, $G$, and $D$ (respectively), this is the quantity $|Pr_{x \gets U}[D(x)=1] - Pr_{x \gets G}[D(x) = 1]|$. Note that the minuend is hard to compute (requires enumerating over all strings of the given length), but can be estimated with uniformly random samples. Can you confirm that this is the quantity portrayed in Figure 8? If not (for example, if it is $Pr_{x \gets G}[D(x) = 1]$ that is portrayed), the graph might falsely claim that the distinguishing ability increases with context length, without any real gain in advantage.
- Am I missing any key experimental result presented in the body of the paper? How do the experiments support the claim that ICL is Bayesian model selection? (How would a different outcome of the same experiments falsify this claim?)

---

> ### Author Response · Authors · 2023-11-19
>
> **(Part 1/4)**
>
> Thank you for the helpful suggestions and feedback! We are very glad that you see this work as “important”, “creative”, “original”, “interdisciplinary”, and “well-written”. We respond to your specific comments below, beginning with some clarifications on two key points of our work that *may* have been misinterpreted.
>
> **Points for further clarification**:
>
> 1. The transitions we observe, for example in Randomness Judgment tasks, are (a) sharp and sudden, and (b) move from a state of strongly following one response pattern (“Random”) to strongly following a very different response pattern (“Non-Random”). Both points are critical to understanding the significance of this result and we have added further clarification in Section 2 and Figure 2 to explain this point, contrasting model selection with model averaging interpretations of ICL, and how our experiments lend support for the former. To our knowledge, our work is novel in being the first example of such a dramatic phase change, or S-shaped curve, in ICL or few-shot learning with an LLM. We observe these in-context learning dynamics in our formal concept learning experiments across two separate tasks, with separate analyses: flip sequence Generation, and randomness Judgment. The implications of this phenomenon may be significant if such in-context phase changes are found in other few-shot learning domains.
>
> 2. None of our experiments involved prompting an LLM with two strings simultaneously, i.e., we never task an LLM with determining which string was generated by a given source. Some of the questions in your review suggest that there may have been a misinterpretation of our experimental protocol, though it is possible that this is merely an issue with phrasing. *To clarify:* our experiments tested LLM (a) inferences about the *source* that generated a sequence of flips (i.e. was the source “Random” or “Non-Random”?), and (b) next-token completions for a sequence of flips, e.g. “\[Heads, Tails, Heads, \_\_\_\_”   (see Section 4 for further details).
>
>
> > For the distinguishing experiments, …. The authors find that each language has a sharp phase shift …  the likelihood that GPT predicts the context to be determinstically-generated increases from ~0.5 (a random guess) at n=3, to ~0.8 at n=4.
>
> We would like to clarify some important details of our results. The specific values you list seem to be inaccurate (.5, .8) based on Figures 4, 5, which is related to our clarification point (1) above. The reason this experiment is interesting is because the likelihood sharply changes from high confidence in a random process to high confidence in a non-random algorithm. That this change is *sharp*, and the LLM transitions *from strongly following one pattern to strongly following another*, are both essential parts of why this result is significant. In addition, the phrasing in the second half of the first sentence implies that we tested GPT with uniformly random sequences $x$ in the Randomness Judgment condition. Following Reviewer xnie’s comments (a different reviewer), our updated version of the paper includes such an analysis, however, our original pdf did not include this.
>
>
>
> > Usage of the term "pseudorandomness"
>
> Thank you for this useful input! We were not aware of this being established terminology in computer science, and we have tweaked our paper to remove “pseudorandom” to avoid this confusion. Our intended meaning is, as you inferred, “seemingly random”, or “subjectively random” as in the cognitive science literature.
>
>
>
> *(continued below)*

---

> > ### Author Response · Authors · 2023-11-19
> >
> > **(Part 2/4)**
> >
> >
> >
> > > What do the experimental results actually mean?
> > >
> > > A significant contribution of this paper is in demonstrating how subjective randomness can be used to experiment on LLMs. However, I am not sure what insight can be garnered from the particular experiments presented in this paper. My understanding of the results from the experiments presented in the body of the paper is:
> > >
> > > 1. GPT does not generate uniformly random numbers; in particular, it exhibits the gambler's fallacy.
> > > 2. As context length increases, GPT can more confidently distinguish uniformy random strings from extremely simple, deterministically-generated strings (such as (01)^n).
> >
> > Thank you for this question. We first suggest that your second point might relate to our above note “points for further clarification (1 & 2)”.
> >
> > Next, we would like to highlight some core contributions of this work, adding to the summaries you have provided.
> >
> > 1. GPT not only exhibits the gambler’s fallacy, but it does so in a manner that cannot be simply attributed to memorizing training data, suggesting ICL is invoking a latent algorithm in GPT, which generates locally balanced binary sequences, similar to our window average model. This behavior is also controllable, with GPT sequence means approximating different probabilities P(Tails), but still exhibiting the same gambler’s fallacy pattern which cannot be explained by simple memorization. It is also worth noting that our experimental setup for subjectively random sequence Generation is fairly sophisticated relative to most cognitive science experiments on subjective randomness, as we have not encountered human experiments analogous to ours, that vary P(Tails) on a weighted coin and demonstrate Gambler’s Fallacy while controlling for how biased the coin is (Figure 8).
> > 2. The interesting part of the results in our Randomness Judgment and formal language learning sequence Generation experiments is that the LLM's behavior *sharply* transitions from *high confidence in “random”* (Judgment task) to suddenly having *high confidence in “non-random”,* or from generating subjectively random sequences (Generation task) to generating deterministic repetitions of patterns such as (01)^n. Our new toy example in Figure 2 is intended to further highlight the important of these learning dynamics, and help the reader understand the meaning of formal concept learning results: that sharp transitions in the predictive distribution suggest ICL acting as model selection.
> >
> >
> > *(continued below)*

---

> > > ### Author Response · Authors · 2023-11-19
> > >
> > > **(Part 3/4)**
> > >
> > >
> > >
> > > > As a suggestion for improvement, I would expand the setup … two axes:
> > > > - Stretch (length of generated string): Rather than testing on deterministically-generated strings (which are NOT pseudorandom in any non-trivial sense of the word), … stretches it into a longer string.
> > > > - Generator complexity: The automaton for detecting the language … significantly more complex programs.
> > >
> > > Thank you for these suggestions. Please see our response below.
> > >
> > > **Generator complexity suggestion**: We have added additional experiments to our submission, testing GPT on Randomness Judgment tasks with significantly longer concepts. Results of these experiments are included in Appendix G. Further, we extended our primary results of formal concept learning on Generation tasks to include additional concepts, as well as longer concepts (Figure 5, Right). In both cases, we observe similar patterns of ICL dynamics, where behavior transitions from stably following a 'random' concept to stably following a 'non-random' concept. As a baseline, this trend does not occur in our Randomness Judgment task when the input sequence is randomly sampled from a Bernoulli distribution (Figure 20). We also observe that this pattern begins to degrade with sufficiently long concepts, where transitions become noisier and require more in-context data.
> > >
> > > **Stretch / Pseudo-randomness  suggestion**: Since we do not have background in computational pseudorandomness, we were unable to devise a relevant experiment for this within the review discussion period. Though we do not have time to complete this for this submission, we would appreciate further details on what a useful experiment with pseudorandomness might look like (perhaps utilizing an algorithm for PRBS [0] with different seeds and/or degrees?). Our updated Appendix G showing Randomness Judgment results with a random baseline for $x$ may be relevant to broader concerns raised in this point.
> > >
> > > **A note on generator complexity:** One essential challenge in defining a space of non-random formal languages is that the language must be able to generate arbitrarily long strings $x$ following a consistent pattern so that we observe ICL dynamics as $|x|$ increases. Consider, for example, the formal language $a^n b^n a^n$ - if we try to do our formal concept learning task with varying $|x|$, many of the strings for shorter $|x|$ will miss essential parts of the pattern. We also considered using non-deterministic regular languages, for example (000 | 101)+. The second essential challenge of this is that non-deterministic generators for $x$ will generate different samples for a given context length $|x|$. The concept (000 | 101)+ has $2 ^ {N / 3}$ unique strings of length N. If each one of these strings shows a sharp phase change in ICL dynamics as $|x|$ increases, averaging over all these strings may still produce a smooth curve. A similar argument is made by [1], who hypothesize that phase changes may be very common in training deep NNs, but these often go unobserved since test loss typically averages over many samples. We added new discussion for this last point in our resubmitted paper, in Section 2.
> > >
> > > [0] https://en.wikipedia.org/wiki/Pseudorandom_binary_sequence
> > > [1] Michaud, E. J., Liu, Z., Girit, U., & Tegmark, M. (2023). The quantization model of neural scaling.
> > >
> > >
> > >
> > >
> > > > One of the main contributions claimed in page 3 was that the subjective randomness framework will provide convincing evidence that ICL carries out a form of Bayesian model selection. I could not see how the experimental results I listed above support this claim. Please let me know if I am missing a connection here.
> > >
> > > Thank you for this feedback! We have edited Section 2, and added a simple illustrative example in Figure 2, that we hope will help readers see the connection between our claims and our evidence. A brief summary is that Bayesian model averaging between Random and Non-Random hypotheses will give rise to a smooth curve in the posterior predictive distribution $p(y | x)$. On the other hand, model selection will give rise to a sharp discontinuity in the predictive distribution, at the point when the hypothesis posteriors cross and a different single model is selected. *Thus, our results of showing sharp transitions with in-context learning support a theory of ICL as model selection.*
> > >
> > >
> > >
> > >
> > > *(continued below)*

---

> ### Author Response · Authors · 2023-11-19
>
> **(Part 4/4)**
>
>
>
> > Please review all figures in the body of the paper for colorblind-accessability. I am colorblind and had a hard time deciphering Figure 8. There are plenty of guides online on how to make accessible figures, and I am happy to point you to specific ones if that is needed.
>
> Thank you for raising this to our attention. In the next few days, we will work on updating our figures with colorblind-accessible color schemes. Though we have found guides on color blind accessibility, if you have any particular suggestions for useful guides, we would appreciate pointers to them. We want to actively engage in learning how to make inclusive figures.
>
>
>
> > What does "Probability of Concept" mean on the y axis of Figure 8? … (See also the last question in Questions section.) … The mathematical notation in the sum on page 7 was confusing (the righthand side that following "and computing"): I didn't understand what is (yd==1)(i)...
>
> We have updated this section of the paper, at the end of Section 4, to make our methodology with formal concept learning in Generation tasks clearer to the reader. We have updated the y-axis of Figure 8 (now Figure 5) Right to match the notation used in this section, the predictive probability $p(y \in C | x)$ of a generated sequence $y$ matching concept $C$.
>
> This figure is showing results for formal concept learning on a Generation task, as is the tree on the Left of this figure. We believe the reviewer interpreted this figure as results for a Judgment task, as in Figure 4. However, we emphasize that this figure is actually showing sharp transitions in a completely different task from Figure 4, which makes this result more surprising. Instead of answering whether the source of a sequence was y=”Random” or y=”Non-Random”, the model simply keeps generating tokens (e.g. y=”Heads, Tails, Heads, Tails, …”) to continue an input sequence of tokens $x$, and once again we see transition points, where GPT suddenly shifts from generating “random” sequences to deterministically repeating the concept in the input.
>
>
>
>
>
> > Thank you for introducing me to the area of subjective randomness. Can you confirm that my understanding of it is correct, as far as needed to understand this paper? Am I missing a key insight from this area?
>
> We are glad that you appreciate the introduction to subjective randomness–we are really excited about this topic and believe it is fertile ground for future work! We believe there are a few points of clarification that warranted discussion in the interpretation of our claims and results, which we have discussed in our responses above. Please let us know if you still have any questions!
>
>
>
> > When talking about distinguishing ability, the correct quantity to look at is the "advantage" of the distinguisher: … Can you confirm that this is the quantity portrayed in Figure 8? … without any real gain in advantage.
>
> We apologize, but it seems due to a lack of background on the topics mentioned in your comment, we are unable to precisely understand your comment. This makes it difficult for us to confirm whether your interpretation of Figure 8 (now Fig. 5) is correct. It is possible that the question echoes our clarification point (2), since we never presented models with multiple different strings, possibly from different sources, simultaneously. We do note that we have updated the text corresponding with Figure 8 (now Figure 5)---maybe this helps clarify some of the points raised in the question above.
>
>
>
> **Rebuttals Summary**: We hope that our updated submission addresses some of your concerns. We have updated our paper following your suggestions, including a new figure (Figure 2, Appendix D) with an illustrative example to help readers connect our claims with our findings. This example reinforces a key point of our work, that we interpret sharp phase changes in ICL as indicative of model selection. We have also added new experiments, such as those in Appendix G, which may address your point about non-random concept simplicity. After considering these changes, as well as our responses to your individual points above, we hope that you will consider raising your score.

---

> > ### Comment · Reviewer_F7kD · 2023-11-21
> > **Response and score update**
> >
> > # Score update
> > After reading the authors' response, I have decided to modify my scores as follows:
> >
> >  - Presnetation, contribution: 2→3.
> >  - Overall score: 3→6.
> >
> > # Response
> > Thank you for responding to my concerns, editing your paper, and providing welcome clarifications in this discussion thread. My overall response is that the paper has improved significantly over its initial version. While I cannot respond to each comment in the rebuttal individually, I would like to leave the following final notes:
> >
> >  1. The removal of "pseudorandomness" terminology is a noticeable improvement, and makes many of my points above moot. While it would be interesting to study the similarities and differences between pseudorandomness and subjective randomness (as already evidenced by the discussion in this thread), I think that the authors chose wisely to leave this outside the scope of this paper.
> >  2. My concerns with all figures have been addressed.
> >  3. Thank you for adding the experiments on generator complexity in Appendix G. My suggestion of experimenting with stretch is anchored in the theory of pseudorandomness, and would be somewhat contrived to add to the paper in its current form (see item 1 in this comment).

---

### Official Review · Reviewer_xnie · 2023-10-31

**Soundness:** 2 fair
**Presentation:** 3 good
**Contribution:** 2 fair
**Rating:** 6
**Confidence:** 3

**Summary:**

The paper examines how LLMs generate and and detect random binary sequences, drawing analogies to in context learning (ICL) as a type of Bayesian Model selection.  They use the paradigm of Bayesian Model selection to analyze a particular aspect of how LLMs behave.  They make the case that analysis allows investigating cognitive interpretability vs. behavioral benchmarks or mechanistic interpretability.

They examine how subjective randomness can be seen as a type of in-context learning, showing that like humans, LLMs have biases in the types of sequences they generate.   In particular, it’s like under-specified program induction, with the posterior over different types of programs being updated with each new observation.

They also propose a specific model (window average) that accounts for some of the behavior of the generated random sequences better than the true generative process which limits access to more recent history. They go on to show that earlier versions of text-davinci do poorly at sequence generation, while the most recent version does have “subjective randomness behavior”, similar to their proposed model.  They also show that specific sub-sequences occur more frequently than would be expected from a true Bernoilli process.

The authors then go on to show how many repeats of a given seed pattern are required to allow chatGPT to identify the sequence as non-random.

And finally, they also show many repeats of a given “seed” sequence are required to make chatGPT only repeat that final pattern, vs. continuing with a psuedo-random sequence.   This sudden phase shift in how an LLM generates a pattern is argued to be similar to switching hypothesis in Bayesian model averaging

**Strengths:**

## Originality:
Determining whether a binary number sequence is assessed to be random by a human has been pretty extensively studied in the literature, and it is an interesting extension to see how it compares to the behavior of an LLM judging a sequence to be random.  Some parts of their framing are novel, in particular their experiments w.r.t to the effect of seeding with a repeating pattern.  Their Window average model appears to match well the real performance of the LLMs, and they have a clear example of the Gamblers fallacy in this model during the generation.  Overall, their biggest novel contribution is that LLMs can be biased by strong local patterns when trying to generate new random sequences, and that this could be interpreted as Bayesian model selection.

## Quality:
See weaknesses for more discussion about the potential mismatch in what are claiming and what their experiments are actually doing.  With that proviso, the experiments generally appear to be consistent with their claims.  They have an extensive literature review that is well written and easy to follow.  Their setting is well chosen, with potential to give some insights into how LLMs are finding local patterns.

## Clarity:
The paper is sufficiently well written, and easy enough to follow.

### Other minor comments:
> For this reason, in Fig. 6 we show results for P (Tails) = 51%.
It would be clearer to also update Figure 6 to show that it is 51%, as opposed to only mentioning it in the text.

> In randomness Generation tasks, we asses concept learning according
Should be “assess”

**Weaknesses:**

The analogy of generating random sequences to in context learning is limited in that the model depends on LLMs getting confused when they are told they are provided with a random sequence, but in fact the “starting” sequence has strong local patterns. Admittedly, this is a short-coming of the LLM as it shouldn’t be “distracted” by this local pattern.  However, this artifact may go away when the base model is improved, and not “distracted” by the local misleading start, which could limit the usefulness of that part of the analysis.

In other words, it can be argued that the analysis is simply examining a failure model of an LLM, namely in a regime where local patterns start to dominate the predictive behavior of the system and it “forgets” the prompt. That doesn’t mean the transformer is doing any sort ICL - it’s just dominated by the local signal.

For example, if you put in any prompt, then a long enough sequence of 001001, it may ignore the prompt and start predicting 001001… as it’s “forgotten” about the prompt due to the strong local predictable signal.     This makes the connection to in context learning more tenuous due to the potential disconnect between the prompt and the task, which may not happen in larger/newer/more refined models or in a more complicated ICL problem.

One way to address mismatch could be the prompt being explicit about it being an ICL problem.  For example:

“”“Continue the following potentially random sequence: <...>”””

Then measuring the success rate of the continuation as you know the underlying generate model.   This would remove the mismatch between the prompt and the model.

Additionally, experiments for determining whether a sequence is random should include some truly random baselines.  For example, Figure 4 has no truly random baseline.  Hence, it is not clear from the current results if the model incorrectly assumes that all sequences are not random given enough observations.  This could be due to limitations in the attention or number of layers, for example, or biases in the training data.  What is the pattern where the LLM starts to fail?

**Questions:**

> ChatGPT generates higher complexity sequences than chance. We speculate that this phenomenon might explained by a cognitive model that avoids sampling with replacement.

I’m unsure what this means - can you elaborate what you mean by higher complexity in this context?

What is the tokenizer doing?  Given that the whole paper is based around heads/tails sequences, how are they being represented?  Are there equal number of tokens being used for Heads and Tails?  Are tokens straddling between Heads/Tails at all, or on the punctuation, which may induce some of the observed patterns?

Correspondingly, does it matter how the random sequence is represented?  Heads/tails might have their own strong biases not represent in random token choices as there are many examples in the training data of these tokens in various non-random contexts.

Why do the prompts specify generating 1000 tokens, but only ever the first 50 are used?  Is there a particular reason for that mismatch?

---

> ### Author Response · Authors · 2023-11-19
>
> **(Part 1/2)**
>
> Thank you for the helpful suggestions and feedback! We appreciate that you find our paper “well written” and “easy to follow”, and our experimental setting to be “well chosen” and “novel”. We are also glad that you appreciated our claim that phase changes in ICL suggest model selection, which we sincerely believe is a fertile avenue for future research!
>
>
> > For this reason, in Fig. 6 we show results for P (Tails) = 51%. It would be clearer to also update Figure 6 to show that it is 51%, as opposed to only mentioning it in the text.
>
> Good point, we have updated Figure 8 (previously 6) to state this.
>
>
> > “In other words … it’s just dominated by the local signal. For example, if you put in any prompt, then a long enough sequence of 001001, it may ignore the prompt and start predicting 001001… as it’s “forgotten” about the prompt due to the strong local predictable signal. … One way to address mismatch could be the prompt being explicit about it being an ICL problem. For example: ‘Continue the following potentially random sequence: <...>’”
>
> These are good questions, and we have run new experiments to address these points.
>
> **Experiment 1 (on prompt specification)**: We re-ran the formal concept learning task in randomness generation with a different prompt, analogous to what you have suggested: “Generate a sequence of 1000 samples that may be from a fair coin with no correlation, or from some non-random algorithm”. We observed similar results, with stable transitions from random behavior and deterministic concept learning, and we have updated Figure 5 and accompanying text accordingly (note that this updated figure also includes 4-length concepts, which the previous version did not).
>
> **Experiment 2 (on local bias)**: In Appendix E , we provide a new experiment to demonstrate that sharp phase changes in ICL as a function of additional context cannot be simply attributed to *local bias* leading the model to copy the pattern of the last few tokens of text, regardless of the overall task. Specifically, we append a sequence of “Heads, Tails, …” (i.e. the concept (HT)+ ) to the end of a zero-shot prompt for a widely used LLM evaluation dataset, MMLU, testing 5 different subjects. In this experiment, we show that GPT shows a strong task bias, rather than merely being “forgetting” the prompt and being “dominated by the local signal”. Even after appending *hundreds* of distractor tokens following a repeating pattern “Heads, Tails, Heads, …” to an MMLU prompt, *GPT reliably converges back to answering the MMLU question and maintains a high probability of generating answer tokens* (“A”, “B”, “C”, “D”, since MMLU is multiple choice) over the course of in-context learning. We provide further discussion in Appendix E.
>
>
> > “Additionally, experiments for determining whether a sequence is random should include some truly random baselines. … What is the pattern where the LLM starts to fail?”
>
> Thank you for this suggestion! To address this point, we have added random baselines for Randomness Judgment to Appendix G as well as accompanying discussion. ICL dynamics for random sequences vary widely, and do not show the characteristic sharp transitions we find for our non-random concepts, further corroborating our perspective on ICL as model selection instead of model averaging.
>
> *(continued below)*

---

> ### Author Response · Authors · 2023-11-19
>
> **(Part 2/2)**
>
> > “ChatGPT generates higher complexity sequences than chance. …” I’m unsure what this means - can you elaborate what you mean by higher complexity in this context?
>
> We interpreted your comment as a request for clarification about this remark, please let us know if this was not your intent and we’ll follow up. Accordingly, we have updated this line in the paper to clarify this point: “... if the LLM makes a sequence more subjectively random by avoiding repeating sub-sequences, like a human might, then it will produce sequences with higher-than-chance complexity overall.”.
>
>
> > What is the tokenizer doing? Given that the whole paper is based around heads/tails sequences, how are they being represented? … prompts specify generating 1000 tokens, but only ever the first 50 are used? Is there a particular reason for that mismatch?
>
> We appreciate your questions, as these prompting details were intentional, and we have updated Appendix B to better clarify them. We chose the format “Heads, Tails, …” specifically to avoid the alternate interpretation you describe, of some patterns merely being tokenization artifacts, since flips with shorter string formats, such as 0s and 1s with no comma separating them, often get chunked into different tokens (“01”, “000”, etc.). We specify 1000 tokens to avoid issues with GPT generating too few tokens, where for example if the *'50 samples'* is specified, less than 50 flips will be generated in some cases. In the future, we hope to explore how different token representations influence results, similar to [0], but with black-box in-context learning behavior.
>
> [0] Wei et al. (2023). Larger language models do in-context learning differently.
>
>
> **Rebuttals summary**: Thank you again for your feedback! In response to your point about concept learning in generation tasks, we updated the prompt for this task and re-ran the corresponding experiment, with relatively similar results. Diving deeper into this point, we conducted a new experiment to test whether, as you suggest, GPT is merely forgetting the prompt when it begins repeating the deterministic pattern of one of our concepts. Our results suggest *not*; instead, GPT has a strong bias to stay on-task for an unrelated task (MMLU benchmark questions), even after many “Heads, Tails, …” tokens are appended to the prompt. We also added a Randomness Judgment baseline using sequences sampled from a Bernoulli distribution (Figure 20), following your suggestion, and found these in-context learning curves to have very different patterns than our repeating concepts. Besides these changes, we have also made smaller tweaks following some of your particular suggestions, e.g. adding tokenization details to Appendix B. We hope these updates address your concerns and hope that you will consider raising your score to support the acceptance of our work.

---

> > ### Comment · Reviewer_xnie · 2023-11-21
> >
> > Thank you for your changes and clarifications.  I've increased the score to 6 based on the updates and the rebuttal.
> >
> >
> >
> > ### Prompt mismatch.
> > Thank you for running the additional experiments with the effect of the prompt mismatch and the local bias effect. I wanted to note that the rewriting of the prompt slightly changes the "ownership" of the provided sequence of being produced from the model vs. provided by an external source.  As in, the suggestion was "provided sequence, continue it < start>
> > , continuation:", vs. "generate a sequence either random or not: <start>". It could, in theory, treat a provided sequence differently than a sequence it is told it has itself generated.
> >
> > As an extension, it would be interesting to see what patterns it chooses to generate without guidance from the prompt as well.
> >
> > The local bias variation is also interesting to see, and might be related to transformers preferring to ignore the middle of the attention window - see [1].
> >
> > [1] Lost in the Middle: How Language Models Use Long Contexts Nelson F. Liu, Kevin Lin, John Hewitt, Ashwin Paranjape, Michele Bevilacqua, Fabio Petroni, Percy Liang
> >
> > ### Random baselines
> > The current plotting is a bit hard to follow.  It would be useful to add a plot for mean + std of P(random|random sequence), mean+std of P(random |length 3,4,5,... concept), where for longer concept they could be subsampling to keep compute costs down.
> >
> > ### Tokenization:
> > Thanks for the added section on tokenization.  I was in particular curious if there was a difference between heads and tails (2 vs 3 tokens for example), or higher likelihood to be joined with the separator/more likely to be split up differently as token are redundant.  From examining the diagram in the appendix, it appears that Heads is a single token while Tails is a pair of tokens, not including punctation.  It doesn't seem that bias has any noticable effect downstream, but it would be more informative to more clearly show this.

---

### Official Review · Reviewer_272r · 2023-11-01

**Soundness:** 3 good
**Presentation:** 2 fair
**Contribution:** 2 fair
**Rating:** 6
**Confidence:** 3

**Summary:**

The paper demonstrates that models of certain size exhibit what they call subjective randomness. That is, a sequence that looks random as there is no discernible simple program that can generate such sequence. They also find that model beyond a certain size can distinguish simple programs from random programs.

**Strengths:**

The paper is well written and enjoyable to read. The observation made by the authors is novel as far as I can tell and makes connection with a human cognitive bias, which may increase our understanding of those large models. Such understanding if of great interest in the current time for several reason including safety.

**Weaknesses:**

I found the paper to utilize a lot of its space to discuss ideas that seems unrelated to what I understand to be its main message. For example, it discusses in length the idea of "cognitive interpretability" which does not seems novel nor relevant to the idea presented in the paper. While such idea could be touched upon, I don't understand the value of Figure 1 in the dissemination of the idea.

The paper seems a little bit rushed. For example, the caption of Figure 2 seems partly unrelated or outdated to the figure. The caption refers to "green" and "yellow" concepts which cannot be found in the figure itself, making hard to understand what the authors are trying to convey. Figure 5 left seems to be the same figure three times.
Moreover, the appendix has several Figures but they are not accompanied by any text.

The paper utilize closed products which may change making the conclusion made in the paper non-reproducible in the future. The authors are encouraged to use open models or demonstrate that the observed behaviors cannot be attained with the currently available open-source models.

**Questions:**

* Does the results change if the authors performed their analysis on an open model instead of a GPT-3.5?
* I don't understand Figure 5 left. How are each graphs different f rom one to another?
* I don't understand Figure 2. The caption points to several  color (e.g. green for $x$), yellow for the concept space embedded in the LLM. But I don't see such colors in the Figure.
* How is the proposed "Cognitive Interpretability" different from qualitative analysis generally done in science and also in the deep learning literature when ones try to understand a model's behaviour? For example, some line of works studied the the attention map of convolutional neural networks when fed certain inputs [0]. This seems to fit more the "Cognitive Interpretability" category than the other interpretability categories.
* Do you see a link between this paper and the emerging line of work on the equivalence of ICL to gradient descent [1]

[0] https://openai.com/research/microscope
[1] https://arxiv.org/abs/2212.07677

---

> ### Author Response · Authors · 2023-11-19
>
> **(Part 1/2)**
>
> Thank you for your useful feedback! We are encouraged that you found our work to be “novel”, “of great interest”, and “enjoyable to read”.
>
>
> > I found the paper to utilize a lot of its space to discuss ideas that seems unrelated to what I understand to be its main message. ... I don't understand the value of Figure 1 in the dissemination of the idea.
>
> Thank you for the suggestion. Given that subjective randomness and Bayesian cognitive modeling are not well known across the ML community, our goal was to describe relevant background and highlight the novelty of this work in analyzing *token-level in-context learning dynamics*. However, we agree that this background can be made more succinct.   We have accordingly  trimmed sections 1 and 2, and moved what was previously figure 1 to Appendix C for further discussion of cognitive interpretability. This reduces the background content to ~1.5 pages now. We have used this space to expand the discussion of our results (see updated text in blue in the latter half of the paper).
>
> > The paper seems a little bit rushed. For example, the caption of Figure 2 seems partly unrelated or outdated to the figure. ... Figure 5 left seems to be the same figure three times.
>
> We appreciate these useful suggestions and have made changes accordingly, though we also stress that our modeling framework and experimental design are thorough and deliberate, and were not created in a hurry. Specifically, we note that in Figure 6 right (previously Fig. 5 left), though the graph structures are the same for all three markov chains, arrow width varies between the three figures. We have amended the caption to emphasize the key takeaway of Fig 6 right - that the top plot has uniform arrows, whereas the middle and bottom figures are irregular but similar to one another. While all figures in the appendix had text accompanying them, a couple figures only had captions and not further discussion. We have updated these sections with further discussion of the results in these figures. For your comment on Figure 1 (previously Fig. 2), we concede that the error you pointed out occurred due to a last minute change in figure aesthetics. We have updated the figure caption to address this and apologize for any confusion.
>
>
> > The paper utilize closed products ... authors are encouraged to use open models or demonstrate that the observed behaviors cannot be attained with the currently available open-source models.
>
> This is a very reasonable point, and we agree that this is important for future replicability. We have accordingly begun conducting Randomness Generation experiments with Llama 2 (7b and 13b) using the hugging face API. Because of limited time, we have not yet included these in the paper, though we are trying our best to add some results before the end of the review discussion period and to add further replications of our other experiments thereafter. Due to resource constraints, it is unlikely that we will be able to run Llama 2 70b during this time period, however, in the future we may be able to acquire computational resources to run the largest open-source LLMs on our tasks.
>
> *(continued below)*

---

> ### Author Response · Authors · 2023-11-19
>
> **(Part 2/2)**
>
> > How is the proposed "Cognitive Interpretability" different from ... For example, some line of works studied the the attention map of convolutional neural networks when fed certain inputs [0].
>
> Thank you for this comment, which prompted us to make a standalone background section in the appendix (see Appendix C.) Therein, we further define Cognitive Interpretability and describe its significance. In brief, this is different from [0] in two important ways. First, *we match cognitive models with behavioral observations of deep NNs*. Our window average model, as well as our framework of ICL as model selection (or, equivalently, hypothesis search) are inspired by models and theories from cognitive science. While [0] makes anecdotal analogies between visual circuits in artificial NNs and circuits in the human visual cortex, we provide formal models and compare behavior and learning dynamics between an LLM and these models. Second, we analyze *in-context learning dynamics*, where behavior changes as a function of addition context data, i.e., with increasing $x$ in length. This is similar to how learning dynamics are studied and modeled in cognitive science, and LLMs present new opportunities for understanding the structure of behavior underlying ICL capabilities. Unlike [0], we do not examine hidden units in deep NNs, we analyze input-output *dynamics*.
>
> > Do you see a link between this paper and the emerging line of work on the equivalence of ICL to gradient descent [1]
>
> Indeed! Our original draft cites [1] in the second paragraph and discusses this perspective. To emphasize this relationship further, we have also updated our paper to emphasize our framework of ICL as discrete hypothesis search. We hope these updates will further clarify this point, so readers immediately see the connection of our work to previous work such as [1]. A key takeaway is that, unlike most prior work viewing ICL as gradient descent or as few-shot learning with loss steadily decreasing with more samples (“shots”), we find sharp phase changes in ICL, analogous to phase changes observed when training NNs.
>
>
>
> **Rebuttals summary**: Thank you again for your feedback. Based on your suggestions, we have shortened the background sections of the paper and moved discussion of Cognitive Interpretability to Appendix C, hence improving the paper’s presentation and addressing your concern about background details. We also agree with your point that open-source LLMs are important for replicability and we are developing a pipeline to replicate our experiments with open-source LLMs. We hope to add initial randomness Generation results with LLaMa 2 by the end of the review period, but are limited by time and resource constraints currently. We promise these results will be included in the final version if we’re unable to meet the deadline right now. We also emphasize we have made further edits to the submission, including additional experiments and a new explanatory figure. In considering these and our responses to individual points you raised above, we hope that you will consider increasing your score to support the acceptance of our paper.

---

> ### Comment · Reviewer_272r · 2023-11-22
> **Post-rebuttal comment**
>
> I thank the authors for their careful response. While I am raising my score to a 6, I encourage the authors to conclude their experiment on an open model and report their finding whether positive or negative.

---

### Meta-Review · Area_Chair_mxT2 · 2023-12-12

**Metareview:**

An interesting contribution bridging cognitive science and machine learning to study subjective randomness in large language models. The reviewers had reservations about the experimental evidence that were partially overcome during the discussion period.

**Justification For Why Not Higher Score:**

modest contribution to an established line of work (biases in large language models)

**Justification For Why Not Lower Score:**

interesting results from a nicely-designed study

---

### Decision · Program_Chairs · 2024-01-16

Accept (poster)